# How well does Digital Soil Mapping represent soil geography? An investigation from the USA

David G. Rossiter[1,2], Laura Poggio[1], Dylan Beaudette[3], and Zamir Libohova[4]

[1]ISRIC-World Soil Information, Postbus 353, Wageningen 6700 AJ, NL
[2]Section of Soil & Crop Sciences, New York State College of Agriculture and Life Sciences, 233 Emerson Hall, Cornell University, Ithaca NY 14853 USA
[3]USDA-NRCS Soil and Plant Science Division, 19777 Greenley Rd., Sonora, CA, 95370 USA
[4]USDA-ARS, Dale Bumpers Small Farms Research Center, 6883 South State Highway 23, Booneville, AR 72927 USA

**Correspondence:** David G. Rossiter (david.rossiter@isric.org)

**Abstract.** We present methods to evaluate the spatial patterns of the geographic distribution of soil properties in the USA, as shown in gridded maps produced by Digital Soil Mapping (DSM) at global (SoilGrids v2), national (Soil Properties and Class 100m Grids of the USA), and regional (POLARIS soil properties) scales, and compare them to spatial patterns known from detailed field surveys (gNATSGO and gSSURGO). The methods are illustrated with an example: topsoil pH for an area in central New York State. A companion report examines other areas, soil properties, and depth intervals. A set of R Markdown scripts is referenced so that readers can apply the analysis for areas of their interest. For the test case we discover and discuss substantial discrepancies between DSM products, as well as large differences between the DSM products and legacy field surveys. These differences are in whole-map statistics, visually-identifiable landscape features, level of detail, range and strength of spatial autocorrelation, landscape metrics (Shannon diversity and evenness, shape, aggregation, mean fractal dimension, co-occurence vectors), and spatial patterns of property maps classified by histogram equalization. Histograms and variogram analysis revealed the smoothing effect of machine-learning models. Property class maps made by histogram equalization were substantially different, but there was no consistent trend in their landscape metrics. The model using only national points and covariates was not substantially different from the global model, and in some cases introduced artefacts from a lithology covariate. Uncertainty (5–95% confidence intervals) provided by SoilGrids and POLARIS were unrealistically wide compared to gNATSGO/gSSURGO low and high estimated values and show substantially different spatial patterns. We discuss the potential use of the DSM products as a (partial) replacement for field-based soil surveys. There is no substitute for actually examining and interpreting the soil-landscape relation, but despite the issues revealed in this study, DSM can be an important aid to the soil surveyor.

## Contents

# 1 Introduction

Digital Soil Mapping has been defined (under the earlier term "Predictive Soil Mapping") as "the development of a numerical or statistical model of the relationship among environmental variables and soil properties, which is then applied to a geographic data base to create a predictive map" (Scull et al., 2003). Since the seminal paper of McBratney et al. (2003), recently reviewed by Minasny and McBratney (2016), DSM has been widely-applied from the field to global levels. This is in contrast to what we here call "traditional" soil survey, in which the soil surveyor develops a mental model of the soil geography (Hudson, 1992) by interpreting the landscape with the aid of airphotos, purposive transects, and detailed profile descriptions at locations thought to represent the central concepts of the soil classes present in the study area (Soil Survey Division Staff, 2017).

A principal attraction of DSM is that it produces consistent, geometrically-correct and reproducible gridded maps over large areas, given training data ("point" observations of soil classes, properties or conditions), a set of environmental covariates covering the entire area to be mapped at some fixed grid resolution, and a set of algorithms implemented in computer code. This removes the need for expertise in discovering and interpreting the soil-landscape relations, also known as the "paradigm" of soil survey (Hudson, 1992), which is vital for traditional soil survey and difficult to acquire and harmonize among surveyors. However, expertise in soil-landscape relations is still needed to ensure that DSM outputs are reasonable, and to discover reasons for any discrepancies.

Further, it may be that fewer locations can be visited in order to develop reliable models, as compared to traditional survey techniques. If the relation with covariates is strong, and locations representative of the entire covariate feature space are included in the training set, it may be possible to map large areas from relatively few field observations. This corresponds to the "homosoil" concept (Mallavan et al., 2010): identical environmental conditions (as represented by covariates) should result in the same soils. Maps made by DSM can include areas that are not accessible to field mappers because of permissions or difficult access, if the available training data cover the covariate space of the inaccessible area. However, DSM requires sufficient sampling density to cover the full covariate space, since most DSM methods do not interpolate or extrapolate in soil property space, and in any case it is inadvisable to predict "too far" from the coverage of the training observations. This has been studied by Meyer and Pebesma, who have developed a method for measuring the distance in both covariate (Meyer and Pebesma, 2020) and geographic (Meyer and Pebesma, 2022) space between prediction locations and the set of training points.

DSM avoids some well-known problems of traditional survey: multiple survey projects over time with inconsistent standards and mapping concepts, inconsistency among mappers, difficulties in objectively identifying boundaries, and indeed the need to identify boundaries. However, traditional soil surveyors and users of their maps are often critical of DSM products, and may not understand how they were made and how they should be used (Arrouays et al., 2020). In the USA there is increasing awareness of, and interest in, DSM products. Here the most important point of contention has to do with DSM resolution (pixel size), which implies a mapping scale, compared to the scale at which differences can be reliably interpreted for user needs. Criticism of DSM products is proportional to the degree to which their implied spatial precision and accuracy is over-sold.

Another benefit of DSM methods is the quantification of uncertainty inherent in various geostatistical and machine-learning approaches (Szatmári and Pásztor, 2018). In traditional mapping, uncertainty is implicitly encoded via mapping scale (which

determines the size of the minimum delineation), map unit purity specification (e.g., complex, association, consociation), and taxonomic precision (e.g., soil series vs. suborder) (Soil Survey Division Staff, 2017).

The success of DSM in reproducing known "point" observations (i.e., pedons described in the field and characterized in the laboratory) is typically reported by evaluation ("validation") statistics based on data splitting or by cross-validation. These evaluations are almost never based on random sampling (Brus et al., 2011), and since the source point datasets are almost always biased towards certain land uses, access constraints or landscape locations, these evaluations carry forward these biases and must be interpreted with caution.

A more serious issue is that point evaluations of DSM products do not consider the spatial pattern of predictions. By contrast, traditional soil surveys produce polygon maps of relatively homogeneous soil bodies (represented as soil map units), with the boundary lines placed at inflection points of maximum change between them (Lagacherie et al., 1996). These maps explicitly show the surveyor's interpretation of the soil landscape as developed from a mental model of the soil-forming processes, and which when viewed as a whole show the pattern of the soil cover. It has long been recognized that the soil cover forms patterns at various scales (Fridland, 1974; Hole and Campbell, 1985), so that the traditional soil mapper attempts to find those patterns expressed at the map design scale. Since DSM predictions are on a grid cell basis, most DSM models have no concept of relatively homogeneous natural soil bodies nor inflection points between them. However, it might be expected that if the values of the DSM covariates representing the soil-forming factors also cluster in a similar pattern to the soil cover, the DSM predictions would also cluster and approximate map units from traditional survey. Convolutional neural networks (e.g., Taghizadeh-Mehrjardi et al., 2020), not represented in the methods compared in this paper, explicitly consider neighbourhoods of various size, but not explicitly connectivity. The question is thus to what degree DSM products represent the actual soil landscape spatial pattern and, more importantly, the underlying pedogenetic and geomorphic processes.

DSM maps are most commonly produced at grid cell resolutions from 1 km to 30 m, and even to $< 10$ m for precision agriculture applications. Environmental covariates are available at these resolution, so that DSM products at high resolutions can show fine details that can not be presented at the design scale of polygon maps made by traditional methods. These have minimum legible delineations (MLD) of 0.25 cm$^2$ (Vink, 1975) or 0.40 cm$^2$ (Forbes et al., 1982) on the published map, multiplied by the scale factor. For example, a polygon map at 1:24 000, typical of USA traditional soil survey, can represent spatial patterns of 1.44 (Vink) to 2.3 (Forbes) ha minimum-size polygons. The Forbes et al. (1982) criteria have been incorporated into NRCS soil survey standards (Schoeneberger et al., 2012; Soil Survey Division Staff, 2017). These correspond to single grid cell resolutions of 240 to 384 m, coarser than higher-resolution DSM products from (30 to 100 m). But the question remains whether this implied fine detail represents true differences or artefacts of the mapping process – in other words, should the DSM map unit trust the apparent differences between adjacent grid cells, or are some or most of these differences due to artefacts ("noise") of the DSM process? Further, there is the question of how well the medium-resolution products (e.g., 250 m) represent the soil landscape at regional extent.

The objective of this study is to present methods with which to evaluate the landscape and detailed level spatial patterns of DSM maps. These maps have been developed for global, national, or regional spatial extents. These patterns are compared with digital soil maps based on polygon maps produced by traditional soil survey, using field study and expert soil-landscape

analysis. We chose the USA as a study area because of the availability of field-based soil surveys at 1:12 000 to 1:24 000 design scale, linked to detailed descriptions of modal soil profiles, available as a seamless digital product. These comparisons may be useful in the context of current plans (Thompson et al., 2020) for updating and completing the USA soil survey using DSM methods and GlobalSoilMap (GSM) specifications (Arrouays et al., 2014). They should also be useful for developing realistic expectations for what DSM can and cannot deliver (Arrouays et al., 2020).

To evaluate DSM methods we apply them to selected test areas and soil properties, and comment on the results. This paper introduces the methods and data sources, and includes an illustrative example (one area, one soil property, one depth interval), in the context of the soil geography of the selected region. A companion ISRIC Report (Rossiter et al., 2021) presents four case studies in diverse soil geographic contexts, each with different soil properties and depth intervals. We encourage readers to apply the methods to their own study areas within the USA and to their soil properties of interest, to evaluate the utility of the several DSM products. For this, we provide our analysis scripts as R Markdown documents (R Studio, 2020); see "Code availability" at the end of this paper.

## 2  Products compared

The products compared in this study differ in their primary data source (soil maps and point observations), their geographic scope, the mapping methods used to make the digital product, their resolution, depths and coordinate reference systems, and how they assess and present uncertainty. We summarize these below; see the journal articles describing each source for details.

### 2.1  General character of the products

The products are of three kinds: (1) digital products based on traditional soil survey without any statistical modelling; (2) DSM products based on traditional soil survey products and enhanced by statistical modelling using environmental covariates; and (3) DSM products based on statistical modelling using training points and environmental covariates. This latter is the most common DSM method worldwide, especially for areas without extensive traditional soil surveys.

The first kind of product is represented by the reference products from the Natural Resources Conservation Service (NRCS) of the United States Department of Agriculture (USDA), based on extensive field survey, airphoto interpretation, thematic maps, and expert evaluation of DEM-derivatives. This is considered to be the most accurate information, despite the occasional presence of artefacts from the overall mapping programme, as explained later in this section. There are two closely-related products from the NRCS.

At the national level, The National Soil Geographic Database **gNATSGO** (NRCS Soils, 2022b) is a composite of the Soil Survey Geographic Database (SSURGO, mostly 1:24,000 scale), and State Soil Geographic Database (STATSGO2, 1:250,000 scale), and the detailed Raster Soil Survey Database (RSS), according to the most detailed product available for all areas of the USA. It is aimed at users who require multi-state or CONUS extent mapping. For each state or equivalent political unit, the SSURGO or STATSGO2 polygon maps of SMU produced by traditional survey have been rasterized to a grid, each cell keyed to a soil map unit (SMU) (NRCS Soils, 2020a). Grid cells link to the best available (i.e., greatest detail STATSGO-SSURGO-

RSS) SMU. The digital products are delivered at 30 and 90 m resolutions for the 48 contiguous states and the federal District of Columbia of the USA (abbreviated CONUS).

At the state level, **gSSURGO** is also available (NRCS Soils, 2022a). This has higher resolution (both 10 m and 30 m) to minimize degradation of the original polygon delineations, and is a direct gridding of SSURGO polygons. It does not use STATSGO2 for infilling nor RSS if available. It thus is a gridded version of the familiar SSURGO product that is used for local applications. gSSURGO is refreshed annually for those users who do not wish to mix STATSGO or the new raster soil surveys into their analysis.

The gridded gNATSGO and gSSURGO maps are derived from the polygons of SSURGO, a representation of those delineated by the field surveyors on stereo-pairs or ortho-photos and subsequently converted to vector digital format by manual digitization. Soil surveys conducted in the last 15 years were compiled using on-screen digitization in a GIS. At boundaries between survey areas, polygon lines at survey limits have been matched during digitizing (D'Avelo and McLeese, 1998). These polygons are organized in soil map units (SMU) with one or more components (soil taxonomic units, STU), usually named for a soil series but more specific than the parent soil series concept. Taxa above the soil series (family or subgroup) are commonly used in soil surveys of national forest land or wilderness areas. Soil series are the lowest level of Soil Taxonomy (Soil Survey Division Staff, 2014) and are described in the Official Series Descriptions (OSD), as modal profiles with a set of ranges for the observed morphology and laboratory measurements. The component STU in a mapped SMU vary in the observed field properties from the OSD modal description, but usually fit within soil series range. The observed field properties of soil component units are utilized for developing a set of interpretations for SSURGO polygon map units. These polygons are available from the NRCS as vector GIS layers (Natural Resources Conservation Service, 2019), and in a convenient format on a geographic background as SoilWeb (California Soil Resource Lab, 2020).

The SMU of the source maps are mappable landscape elements, at the survey design scale. These almost always have multiple component STU, with reported estimated proportion and geomorphic arrangement within the SMU when possible. However the locations of the STU within the SMU are not mapped due to the design scale. The STU are linked to database tables of representative or synthetic soil profiles, with field and laboratory measurements of multiple soil properties, as well as interpretations for soil use. To obtain values for soil properties in a gNATSGO or gSSURGO grid cell, properties of the components of the corresponding SMU are combined by area-weighted averaging. To obtain values at coarser resolutions, weighted-average properties of groups of grid cells are upscaled by averaging.

There are inherent problems with this product. First, since traditional surveys were carried out over a long time period, series names and mapping concepts may differ between adjacent survey areas. Thus SSURGO SMU delineations and linked tabular data represent a progressive data collection and correlation effort spanning nearly 100 years. Therefore, there exist many soil survey vintages, each a snapshot in time, tied to specific land-use assumptions and technological limitations. Systematic, continuous updates to the entire SSURGO database have been made since 2013 and are ongoing. Second, the transfer from unrectified photos to topographic base and the edge matching between survey areas has not always been flawless, and in addition polygons may have been mis-drawn on the original survey (Supplementary Fig. S1). Thus we can not take these primary polygon maps as a completely reliable georeference.

The second kind of product is represented by **POLARIS soil properties** (Chaney et al., 2019) (further **PSP**), the result of harmonizing diverse SSURGO and STATSGO2 polygon data with the DSMART algorithm (Odgers et al., 2014) to produce a probabilistic raster soil class or component map (30 m grid resolution) and then extracting property information from gSSURGO or gNATSGO grid cells (representing polygons), aggregated by component name. Despite the source data, this is not an NRCS product and was developed independently of the NRCS.

There are two products representing the third kind of product, one for the world and one for the continental USA only. This allows us to compare globally- and nationally-consistent products. The global product is **SoilGrids v2.0** (further **SG2**) (ISRIC - World Soil Information, 2020; Poggio et al., 2021), a further development of SoilGrids1km (Hengl et al., 2014) and SoilGrids250m (Hengl et al., 2017). This uses a global point dataset and environmental covariates that cover the entire world (except the high Arctic and Antarctica), and global models. It does not use any information derived from SSURGO or STATSGO map units. Its training points are extracted from the freely-shareable World Soil Information Service (WoSIS) point dataset from ISRIC-World Soil Information (Batjes et al., 2020). These include all profiles in the NCSS Laboratory Characterization Database database. The freely-sharable WoSIS points are augmented by several datasets included in WoSIS that can not be published externally due to restrictions by the original data providers to ISRIC, but which can be used in mapping. In total $\approx 240\,000$ profiles were used in model building.

The continental product is the **Soil Properties and Class 100m Grids of the United States** (further **SPCG**) (Ramcharan et al., 2018), which followed the methodology of Hengl et al. (2017) with the addition of USA-specific covariates, notably parent material and drainage classes extracted from SSURGO or STATSGO2 map units, and only used the CONUS extent of environmental covariates in model building. SPCG is similar to SG2 in that it is primarily based on point observations, but it has a richer source of these than SG2: the NCSS Laboratory Characterization Database (34 183 pedons comprising 213 499 horizons), the National Soil Information System (NASIS), and the Rapid Carbon Assessment (RaCA) dataset (31 215 pedons); this latter only for organic C, total N, and bulk density. It also uses SSURGO map units to derive parent material (87) and drainage (4) classes as CONUS-specific covariates.

## 2.2 Mapping methods

gNATSGO and gSSURGO are based on traditional soil survey, mostly on unrectified airphoto bases until the late 1990's. The many individual survey areas prior to this time have been partially homogenized during a process of digitization and recompilation onto topographic or orthophoto bases during the 1990's (D'Avelo and McLeese, 1998) and are provided as the polygon SSURGO map. In the early 2000's for new surveys and updates a transition was made to on-screen digitization over orthophotos. Field methods are described in successive editions of the Soil Survey Manual (Soil Survey Division Staff, 2017) and the field book for describing and sampling soils (Schoeneberger et al., 2012). Mapping is based on conceptual models of soil-landscape relations developed in each survey area (Hudson, 1992), confirmed by purposive auger and full profile descriptions to characterize map unit composition. Component concepts are refined with any available laboratory characterization data, with (limited) new laboratory characterization performed as needed. Thus SSURGO provides a *local* model of soil-landscape relations, developed in each area from the most significant soil forming factors relevant to that area. SSURGO is progres-

sively updated by field inspection and correlation, as problems are identified by soil surveyors or map users. Since SSURGO is compiled from diverse surveys over many years, in some areas there are artefacts of that survey process (Supplementary Fig. S2).

The three DSM products (SG2, PSP, and SPCG) use a large number of gridded GIS coverages as environmental covariates in their predictive models. These represent soil-forming factors, and include climate, ecology, geology, land use/cover, terrain, vegetation and hydrography (Supplementary Information §4). PSP also uses coarse-resolution ($\approx 2$ km) estimates of U, Th, and K $\gamma$-ray decay products to represent suspected variation in parent material kind and origin.

PSP (Chaney et al., 2019) uses the DSMART disaggregation algorithm (Odgers et al., 2014) to predict the most probable component (STU), along with their probability of occurrence, at each 30 m resolution grid cell, and from the modal soil properties of the component, a probability-weighted aggregation. Disaggregation is the process of examining a coarser-resolution gridded or smaller-scale polygon product which is known to have multiple STU, and identify the locations at a finer grid resolution where these components would be found, should the original survey have been at larger scale. This depends on fine-scale covariates that, in theory, relate to the STU within an SMU. It attempts to deal with the problems caused by multiple surveys over time, inconsistencies among mappers, and poor georeference of SMU boundaries by sampling out of mapped SMU polygons according to declared proportions of map unit components (STU) and using these as pseudo-observations to train DSM models of STU occurrence. PSP does not use any point observations; rather, it samples pseudo-points from gSSURGO or gNATSGO and uses these as training points for the DSMART disaggregation algorithm (see below). The model is trained in overlapping tiles, each containing some set of SSURGO primary surveys, and using covariates covering just the tile. Thus each POLARIS tile is derived from a *local* model in two senses. PSP provides a fine-scale map equivalent to $\approx$1:3 000 design scale, i.e., from 16 to 64 times finer resolution than the original 1:12 000 to 1:24 000 surveys included in SSURGO. An obvious question is whether it is possible to map at this resolution from the SSURGO source, even with the fine-resolution covariates used by DSMART, because of the probabilistic nature of selecting pseudo-points to match with components (STU).

The other two methods are representative of the dominant DSM method as implemented, with some differences in detail, in many countries and for many properties (e.g., Reddy et al., 2021; Liu et al., 2020; Araujo-Carrillo et al., 2021).

SG2 (Poggio et al., 2021) uses random forests implemented in the `ranger` R package, with prior covariate selection by recursive feature elimination and model tuning by cross-validation of model hyperparameters (number of covariates at each tree split, number of trees in the forest). The model is trained for the whole world, not per-country or region, thus it is a *global* model. This is based on the "homosoil" concept (Mallavan et al., 2010): identical environmental conditions anywhere in the world should result in the same soils. Its use in DSM assumes that all soil forming factors are fully specified (i.e., over their whole range and with all their possible interactions) in the model and training set. Due to the uneven distribution of training points in covariate space, as well as portions of covariate space with no observations, this ideal situation is not met. An obvious question is whether or not the additional information from outside the CONUS leads to an improved model for this region.

SPCG (Ramcharan et al., 2018) is an extension of the original SoilGrids approach, but uses an ensemble of two tree-based machine learning methods: random forests (as in the original SoilGrids) and gradient boosting. The model is trained for the

CONUS, not per-region, thus it is a reduced version of the "homosoil" concept. It is a *global* model in the sense of "use all information over a wide area", although this is not the entire globe, as in SG2.

## 2.3 Resolution, depths and coordinate reference systems

About 90% of gSSURGO is derived from polygon maps with a design scale (1:12 000 to 1:24 000, depending on the original survey), which corresponds to MLD 1.44 to 2.3 ha (1:24 000) or 0.38 to 0.575 ha (1:12 000) polygons, depending on the definition of MLD (see above). These correspond to single grid cell resolutions of 240 to 384 m (1:24 000) or 60 to 96 m (1:12 000). gNATSGO includes some areas surveyed at smaller scale (1:250 000). gNATSGO is delivered as gridded coverages at 30 m or 90 m horizontal resolution on an Albers Equal Area projection covering the CONUS, with standard parallels at 29.5°

and 45.5° N and the central meridian at -96° E on the NAD83 datum, which uses the GRS80 ellipsoid. We have used the 30 m resolution product. gSSURGO is delivered as gridded coverages at 10 m or 30 m horizontal resolution on the same CONUS projection. We have used the 30 m resolution product. Property information is provided per horizon or layer, each with depth limits. Thus to produce a prediction for a depth interval these must be aggregated by depth-weighted average by thickness across the depth interval. PSP predicts at 1 arc-second of longitude and latitude resolution, i.e., 0.0002777778° on the WGS84

datum, equivalent to ≈ 32 m latitude, and proportionally smaller longitude depending on latitude. Depth slices are the standards specified by GlobalSoilMap. SPCG predicts at 100 m resolution for seven point depths (0, 5, 15, 30, 60, 100 and 200 cm) in the same projection as gNATSGO and gSSURGO. Predictions are means of a depth interval. SG2 predicts at 250 m resolution for the standard depth intervals specified by GlobalSoilMap on an equal-area Interrupted Goode Homolosine (IGH) projection on the WGS84 datum (Moreira de Sousa et al., 2019). Depth slice predictions are in fact point predictions at the centre of

the depth interval, considered to represent that interval. The Supplementary Information §3 explains how these products are accessed and made compatible for comparison at regional and local scales.

## 2.4 Uncertainty assessment

SG2 and PSP predict the 5% and 95% quantiles of the distribution of predictions. SG2 uses Quantile Regression Forests (QRF) (Meinshausen, 2006), whereas PSP's uncertainty estimates are based on property data available for each STU predicted by

280 POLARIS. The profile property data are used to create a depth-harmonized profile with uncertainty for each standard depth interval.

These uncertainty limits are specified by the GlobalSoilMap consortium (Arrouays et al., 2014), defined as "the 90% Prediction Interval (PI) which reports the range of values within which the true value is expected to occur 9 times out of 10 . . . there is no assumption that this prediction interval is necessarily symmetric around the predicted value." (Science Committee, 2012).

gNATSGO and gSSURGO provide "representative", "upper" and "lower" limit values of each property of a STU, per horizon or layer. The National Soil Survey Handbook, §618.2 (United States Department of Agriculture, Natural Resources Conservation Service, n.d.) explains that the representative value approximates the median, but that the quantiles corresponding to the low and high values can be adjusted to the percentiles which best show the spread of the property within an STU. If there are sufficient laboratory data of sampled profiles of the STU in the National Soil Information System (NASIS) (Natural

Resources Conservation Service, n.d.), these are used as the basis for establishing the range. In all cases expert opinion is used to adjust these to represent the range that a map user can expect to find in the field. Thus these are not directly comparable to the results of QRF, but do give some idea of how the field mappers, supported by laboratory observations, conceive of the spread of a property. Note that none of these assessments imply a parametric probability distribution, only ranges of selected quantiles. Libohova et al. (2014) discuss how these estimates can be derived for USA products following the GlobalSoilMap.net specifications.

As pointed out by Arrouays et al. (2020), "[t]he user community requires training in, and experience with, the new digital soil map products, especially about the use of uncertainties". It would be hoped that the uncertainties computed by different methods would be similar.

## 3  Evaluation methods

We compared DSM products at regional (nominal $250$ m grid cells) and local (nominal $30$ m grid cells) levels. We evaluated both qualitatively, i.e., by visual inspection followed by expert interpretation, and numerically, over a $1 \times 1°$ tile, selected based on its diverse soil-forming factors and environments and our familiarity with its soil geography. For the pattern analysis within this area we selected a $0.20 \times 0.20°$ subtile and projected the maps to the UTM18N grid on the WGS84 datum (ESPG code 32618).

To compare maps at the regional resolution ($250$ m), the higher-resolution maps (gSSURGO, PSP, SPCG) were aggregated to the lower resolution by weighted averaging (resampling) of the high-resolution pixels within one low-resolution pixel. Thus there is smoothing inherent in the regional comparisons.

To compare maps at the local resolution, we only included the two products (gSSURGO and PSP) provided at that resolution, along with the global product (SG2) as reference, this latter downscaled by increasing the grid resolution without any attempt to disaggregate within the larger grid cell, over a $0.15 \times 0.15°$ subtile.

### 3.1  Qualitative methods

Qualitative methods for comparing maps rely on expert judgement to identify known soil-geographic patterns and evaluate to what extent they are represented on the gridded maps. The maps are displayed side-by-side along with a map of their pairwise differences. Areas of disagreement are identified and discussed.

The DSM product can be evaluated at selected known points, typically from field observation of test areas: is the "correct" soil type or property predicted? and if not, is the error a reasonable approximation? More interesting are patterns in the DSM product. These can be compared to patterns used in the mental model of traditional soil survey, for example, toposequences and sequences of contrasting parent material.

In both cases (points and patterns) the evaluator may be able to infer which DSM covariates would be needed to improve the map.

## 3.2 Numerical methods – whole map

Numerical methods for comparing gridded maps as a whole include (1) MD: Mean difference (also known as the bias), i.e., the average disagreement between maps; (2) RMSD: root mean squared difference; (3) RMSD adjusted for MD, i.e., the RMSD after subtracting the bias from each prediction. These take the first-listed map as reference and the second as the map to evaluate. They can be normalized by the number of grid cells or total area. In addition, all maps can be compared by their Pearson (linear) correlations. These methods are of limited interpretive value. Their main use is to characterize the bias (MD) over the entire map; they do not reveal where any discrepancies occur. For example, there can be no bias overall but a large difference in the amount and values of higher and lower differences. This will be reflected in the RMSD, although not shown on a map.

## 3.3 Numerical methods – spatial continuity

Soil properties are usually spatially-correlated: we expect similar values of properties in nearby grid cells. The degree of local spatial continuity can be assessed by the variogram computed over local neighbourhoods of the gridded map. We computed and modelled the variogram within a local neighbourhood and automatically fit with an exponential model, using the `fit.variogram` function of the `gstat` R package (Pebesma, 2004). Spatial structure is characterized by the range, proportional nugget and structural sill of the fitted variogram model. The range shows the radius over which the selected property has spatial correlation. The proportional nugget shows the variability at the prediction point at the centre of a grid cell, at a scale shorter than the grid spacing. The structural sill shows the overall variability within the range. These metrics show differences in spatial continuity (range), total variability (total sill) and short-range unexplained variability (proportional nugget) between maps.

## 3.4 Numerical methods – patterns

Numerical methods for comparing patterns include: (1) the "V-measure" method (§3.4.1) (Nowosad and Stepinski, 2018) implemented in the `sabre` "Spatial Association Between REgionalizations" R package (Nowosad, 2020); and (2) landscape-level metrics (§3.4.2) (Uuemaa et al., 2013) as used in ecology and derived from the FRAGSTATS computer program (McGarigal et al., 2012), implemented in the `landscapemetrics` R package (Hesselbarth et al., 2019). These include Shannon diversity and evenness, landscape shape index, and fractal dimension. Although the ecological relevance of FRAGSTATS metrics have been criticized (Kupfer, 2012), here we use them to characterize spatial patterns of soil properties, not as inputs to landscape ecology models. Most of the metrics used here have also been used by Pindral et al. (2020) in a study of urban pedodiversity.

These methods must be applied to classified maps, so the continuous soil property maps must first be classified into ranges before analysis. Different choices of class limits and widths will result in different values of these measures. A somewhat objective method to choose classes is histogram equalization. The analyst determines the number of classes, and equal numbers of grid cells are in each class. To compare maps, the combined values of all maps are used to construct the histogram. For the V-measure the gridded maps must be polygonized.

### 3.4.1 V-measure

The V-measure metrics compare different spatial partitions of the same domain, in this case, maps with classified soil properties. The intent is to reveal how similar are these partitions. Two maps could have the same total areas of each class, and even the same number of polygons within each class, and even the same size distribution of these polygons, and yet be completely different in how they partition space into classes.

The polygons of a classified map are termed *regions* of a *regionalization* in the first (reference) map and *zones* of a *partition* in the second map. These are intersected to produce *segment* polygons of the combined map, which are labelled with both zone and region classes. These polygons are then used to compute two metrics of the map to be evaluated (1) *homogeneity* and (2) *completeness*, both with respect to the regionalization of the reference map.

The *homogeneity* of the second map is a measure of the variance of the regions within a zone, normalized by the variance of the regions in the entire domain of the first map. These variances are computed by the Shannon entropy based on areas of the segments. If the variance of the regions within the zones is small, the partition is relatively homogeneous with respect to the regionalization. A perfectly homogeneous partition (with value 1) is when each zone of the second map is within a single region of the reference map. In this case each zone has only one reference class. A perfectly inhomogeneous partition (with value 0) is when each zone has the same composition of regions as the entire domain of the first map, i.e., the second map's partition (to be evaluated) is essentially random with respect to the first map's regionalization.

*Completeness* of the second map is the inverse of homogeneity: it assesses the variance of the zones within a region, normalized by the variance of the zones in the entire domain of the second map. It evaluates the homogeneity of regions with respect to zones, and shows how well the regionalization of the reference map fits inside the partition of the map to be evaluated. A perfectly complete regionalization is when each region of the reference map is entirely within a single zone of the map to be evaluated. In this case a polygon of the reference map will not be split among zones.

These two together are combined into a single measure, the V-measure, as the harmonic mean of homogeneity $h$ and completeness $c$ (Equation 1). This has a range between 0 (no spatial association between the maps) and 1 (perfect association). Obviously, we prefer high association between maps produced by DSM and a reference map. We can also assess the agreement of the patterns produced by different DSM methods, selecting one as a reference.

$$V = \frac{h \times c}{h + c} \tag{1}$$

### 3.4.2 Landscape metrics

The landscape metrics applicable to soil maps (as opposed to, e.g., maps of vegetation types) have diverse interpretations. We compare the metrics of two maps to see if they have a similar concept of the soil landscape. The `landscapemetrics` package can compute many FRAGSTAT indices. We chose several that show the landscape-level difference between maps. We did not consider metrics of individual patches, except as they contribute to landscape-level metrics. The algorithms for these can be found in the package code repository (Hesselbarth, 2021); here we present the formulas and their interpretations.

- The **Shannon Diversity Index** shdi (Equation 2), where $p_i$ is the proportion of pixels of class $i = (1 \ldots N)$, characterizing the landscape diversity according to two factors: number of classes and their proportions. It is widely used as a summary diversity measure, although it does not distinguish between the two factors. More classes and/or a more even distribution of proportions lead to a higher landscape diversity. This does not account for spatial contiguity, it just considers the class of each pixel, irrespective of position. In this example the number of classes in each map will be similar, a maximum of eight (the chosen histogram equalization classes, computed over the combined range of all maps), but some maps may lack representatives of the highest or lowest classes, and so will have only seven classes.

$$D = -\sum_{i=1}^{N} p_i \ln p_i \tag{2}$$

- The **Shannon Evenness Index** shei (Equation 3) is a normalization of Shannon Diversity by the maximum diversity possible for the given number of classes (N). It varies from 0 (completely uneven distribution - low landscape diversity) to 1 (all proportions are equal - high landscape diversity). It does not depend on the number of classes, and thus isolates the effect of class proportion.

$$E = \frac{D}{\ln N} \tag{3}$$

- The **Landscape Shape Index** lsi (Equation 4), where $A$ is the total area of the landscape and $E'$ is the total length of edges, including the boundary, quantifies the internal boundary complexity of a landscape tile, with a value of 1 when the landscape consists of a single square patch, increasing without limit as the length of edges within the landscape increases. This metric characterizes the degree of compactness of the contiguous areas of the classes.

$$\text{LSI} = \frac{0.25 E'}{\sqrt{A}} \tag{4}$$

- The **Landscape Aggregation Index** lai (Equation 5), where $g_{ii}$ is the number of like adjacencies, $(max - g_{ii})$ is the classwise maximum possible number of like adjacencies of class $i$ (i.e., if all pixels in the class were in one cluster), and $P_i$ is the proportion of landscape comprised of class $i$, to weight the index by class prevalence. Thus lai equals the number of like adjacencies divided by the theoretical maximum possible number of like adjacencies, summed over each class and over the entire landscape. It ranges from 0 for maximally disaggregated to 100 for maximally aggregated landscapes. This metric characterizes how dispersed are the classes.

$$\text{AI} = \left[ \sum_{i=1}^{m} \left( \frac{g_{ii}}{max - g_{ii}} \right) P_i \right] (100) \tag{5}$$

- The **Mean Fractal Dimension** `frac_mn` characterizes the complexity of the landscape as the mean of the fractal dimension of all patches in the landscape. It approaches 1 if all patches are square, and 2 if all patches are irregular. It is scale-independent. The patch-level fractal dimensions are computed from the patch perimeters $p_{ij}$ in linear units and areas $a_{ij}$ in square units; these are then averaged to obtain `frac_mm`.

$$\text{FRAC} = \frac{2 * \ln *(0.25 * p_{ij})}{\ln a_{ij}} \tag{6}$$

- The **Co-occurrence vector** `cove` proposed by Nowosad (2021) summarizes the entire adjacency structure of the map and can be used to compare map structures. This is a normalized form of the co-occurrence matrix, which counts all the pairs of the adjacent cells for each category in a local landscape, in the form of a cross-classification matrix. This vector can be considered as a probability vector for the co-occurence of different classes. Co-occurrence vectors of different categorical maps can then be compared by computing the distance between them. Many distance measures are possible; we choose the Jensen-Shannon distance (Equation 7), which computes the entropy $H$ of each probability vector $v_i$ and entropy of their average, and from these the distance in entropy space between them. Increasing values indicate increasing dissimilarity in the adjacency patterns. The computation of `cove` is implemented in the `motif` R package, and the Jensen-Shannon distance in the `philentropy` R package.

$$\text{JSD}(v_1, v_2) = H\left(\frac{v_1 + v_2}{2}\right) - \frac{1}{2}[H(v_1) + H(v_2)] \tag{7}$$

## 3.5 Regional patterns

Regional patterns are at the scale of regional trends such as lithologic units, elevation zones in mountains, and repeating patterns (e.g., basin-and-range, ridge-and-valley). The gNATSGO maps are taken as the reference, although we are well aware that they may not always correspond to ground truth. We then comment on the differences and speculate on the causes, based on our knowledge of the DSM procedures used to make each product and the nature of the soil landscape.

## 3.6 Local patterns

Local patterns are at the scale of geomorphic features such as hillslope catenas, fluvial terraces, outwash fans, valley trains and drumlin fields. This evaluation within the test area for the regional patterns, but examining a smaller area with a distinctive soil-landscape pattern. The gSSURGO maps are taken as the reference. This was evaluated by two methods, as follows.

### 3.6.1 Visual method

We produced ground overlays of key soil properties at selected depth intervals with corresponding KML specifications, and displayed these in Google Earth as semi-transparent overlays, using the original resolution of each product, projected into

WGS84 geographic coordinates as required by Google Earth. These were then compared with gNATSGO maps streamed within Google Earth by SoilWeb Earth (California Soil Resource Lab, 2020). This shows the mapped polygons, labelled with their map unit, and linked to the map unit description, which in turn is linked to the Official Series Descriptions (OSD) (NRCS Soils, 2020b) with complete description of the soil properties modal values and ranges.

### 3.6.2 Quantitative method

This follows the procedures of the regional assessment, except that V-measures are not computed, due to the very fine pattern of classified polygons.

## 4 Example area and soil property

To illustrate the method, we selected one area familiar to the first author, and an important soil property with strong spatial variability and pattern, namely pH in the 0–5 layer. We selected this property because in our experience this is often well-modelled by DSM methods. For example, SG2 had global cross-validation statistics of 0.78 pH median RMSD and a Model Efficiency Coefficient (MEC, the $R^2$ of the 1:1 line actual vs. observed) of 0.67 (Poggio et al., 2021). We select the topmost depth interval because it is most represented by many environmental covariates, especially land cover as well as those derived from remote sensing. Thus the example shown here may be the best case, where we would hope that all mapping methods should provide similar results.

The example area is in central New York State, bounding box (-77 – -76° E), (42–43° N); the subtile for pattern evaluation was (-76.8 – -76.6° E), (42.2–42.4° N), centred at Cayuta NY. The regional geomorphology is described by Bloom (2018). The underlying bedrock is a sedimentary sequence from Ordovician (north) to upper Devonian (south), with a wide variety of sedimentary facies. A strip of the bedrock geology map (New York State Geological Survey, 1970) covering part of the study area is shown in Fig. 1.

The entire area has been glaciated, the portion north of about 42° 15' (Valley Heads terminal moraines) somewhat more recently than the southern portion. A fragment of the surficial geology map (New York State Geological Survey, 1986) is shown in Fig. 2. This shows strongly-expressed features resulting from the most recent glaciation; these are well-known to the traditional soil surveyors. Many glacial features are present and relevant to soil geography: ground moraine, deep glacial troughs with proglacial lake sediments, beach lines, outwash valley trains, kame terraces and hanging deltas. Soil reaction in the northern half is largely controlled by limestone spread by the glacier from outcrops of the Onondaga and Tully limestones (Fig. 1), decreasing to the south.

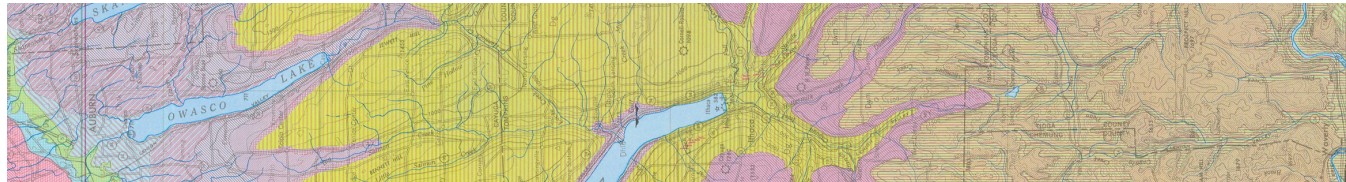

**Figure 1.** Bedrock geology of Central New York State, transect from N–43° N (left) – 42° , centred on -76° 30' E. Orientation N (left) to S (right). Chronological and topographic sequence from Upper Silurian (N) through Upper Devonian (S) sedimentary rocks, notably the Onondaga limestone (green "Don") and Tully limestone (crosshatched red, "Dt"). Source: (New York State Geological Survey, 1970)

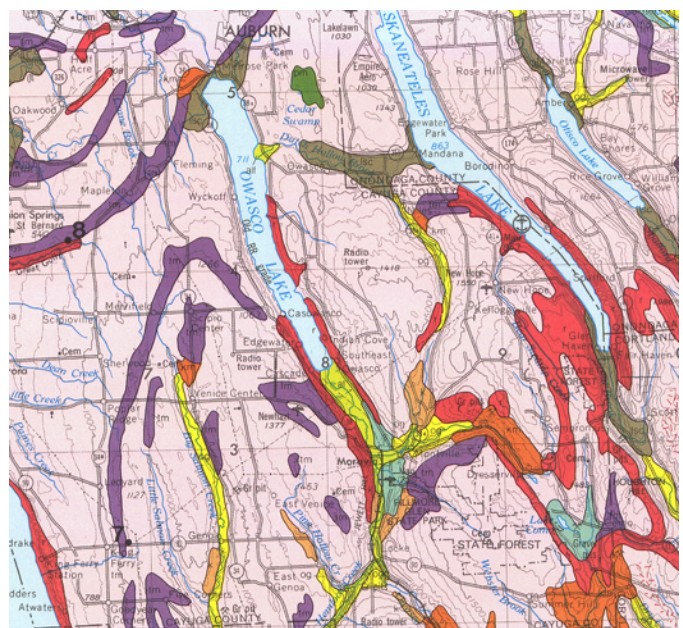

**Figure 2.** Surficial geology of Central New York State near Moravia NY. Legend: ground moraine (pink; if stippled shallow over bedrock), proglacial lakes (brown), organic swamps (dark green), bedrock or very thin soil cover (red), till moraine (purple), kame moraines (orange), lacustrine sand (light green), outwash sand and gravel (yellow). Source: (New York State Geological Survey, 1986)

## 5   Regional spatial patterns

### 5.1   Visual method

Visual inspection of a DSM product over the landscape can be useful to identify anomalies and the degree to which the DSM product captures landscape features. These are over small areas where the soil-landscape relation is known to the evaluator. This can not be part of a systematic evaluation, but can reveal areas of concern or agreement.

As an example, Figure 3 shows SSURGO map units, from a 1965 1:20k design scale soil survey, minimum legible delineation 1.6 ha (Cornell University Geospatial Information Repository (CUGIR), n.d.), draped over a ground overlay of pH (0–5 cm) from SG2, produced by the SoilWeb streaming coverage in Google EarthPro, with a point query showing the SSURGO map unit composition (Figure 4). The map unit is described by its constituent soil series and their estimated proportions. Each series can then be queried for its Official Series Description (OSD) (NRCS Soils, 2020b), which gives a typical profile, a range of properties, and a link to lab data for the series. In this case, the pattern of properties as predicted by SG2 somewhat follows the map unit delineations, but at a much coarser resolution. This is especially evident at the transition from the end moraine (map units beginning with H) and the steep slopes with thin till from local bedrock (map unit LoF).

### 5.2   Regional maps

Table 1 shows the statistical differences between gNATSGO (reference) and the DSM products. All DSM products under-predict topsoil pH with respect to gNATSGO, by about 0.38–0.48 pH units. The RMSD is substantial also, on the order of 0.49–0.67 pH units, somewhat less than this when corrected for bias.

| Product | MD | RMSD | RMSD.Adjusted |
|---------|-------|-------|---------------|
| SG2 | 3.796 | 6.111 | 4.789 |
| PSP | 3.843 | 4.908 | 3.052 |
| SPCG | 4.815 | 6.693 | 4.649 |

**Table 1.** Statistical differences between gNATSGO and DSM products, pHx10, 0–5 cm

Figure 5 shows whole-map histograms. PSP has a bimodal distribution, and predicts few pH values around pH 5.8. This was unexpected, since this value is well-represented in the gNATSGO map. It may be an artifact of a covariate that is influential over a wider area than this tile and results in two regional distributions from contrasting elevation or climate zones. The other distributions are fairly symmetric, although SG2 and SPCG are more even than gNATSGO, which is strongly concentrated near pH 6.2. This shows the smoothing effect of the machine learning models.

Figure 6 shows the pairwise Pearson correlations between the products. The products are overall well-correlated. SG2 and SPCG are very closely correlated, since they use similar mapping methods, despite the additional covariates used by SPCG. PSP and gNATSGO are also closely-correlated. These correlations do not account for bias. They do however show that the maps are similar in their overall pattern as evaluated per-grid cell.

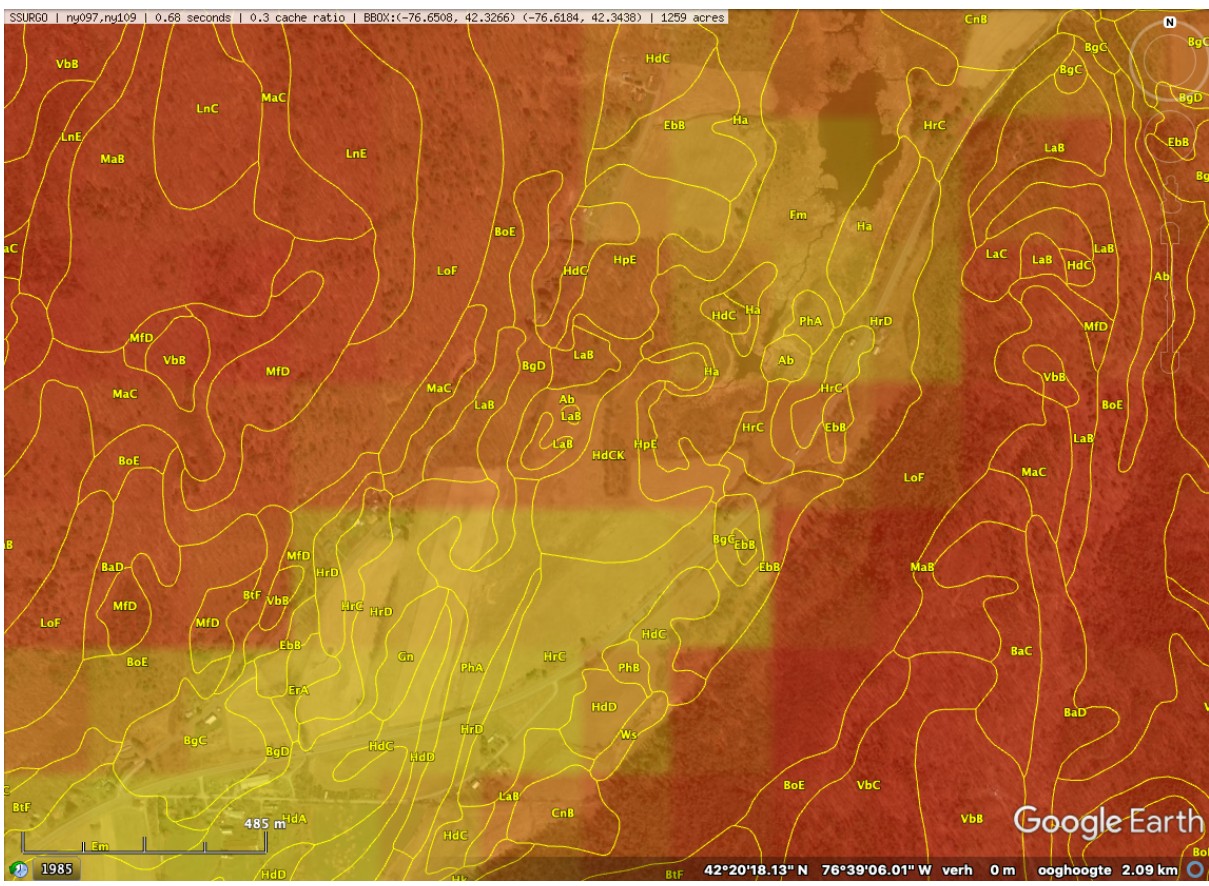

**Figure 3.** SoilWeb view of SSURGO map units and a ground overlay of pH, 0–5 cm, predicted by SG2. Colours are low (red) to yellow (high) pH. Centre at $-76°38'04"\,\mathrm{E}, 42°20'07"\,\mathrm{N}$. Interactive view of SSURGO: https://casoilresource.lawr.ucdavis.edu/gmap/?loc=42.33215,-76.63590,z15

Figure 7 shows gNATSGO (reference) along with the predictions of pH of the PSP products. Figure 8 shows these as difference maps. These figures reveal substantial differences between products. The most obvious is in the detail of the spatial pattern. Despite having been upscaled to regional resolution, gNATSGO shows finer detail than the other products, especially PSP.

These figures also show the spatial distribution of the bias compared to gNATSGO (as reference). SG2 and SPCG underpredict pH in the higher hills in the NE portion of the map, and in the glacio-lacustrine sediments along the lakeshores. The disagreement along the lakeshores is because SG2 and SPCG do not use a surficial geology map, which would be especially useful in recently-glaciated areas such as this. The disagreement with in the higher hills seems to be a direct result of elevation. This is not because of extrapolation in feature space, because at these elevations SG2 also misses the soils derived from Onondaga limestone glacial till towards the southern end of the till plain. SG2 has no information on parent material and uses global models. SPCG has very similar differences, despite using SSURGO-derived parent material as a covariate.

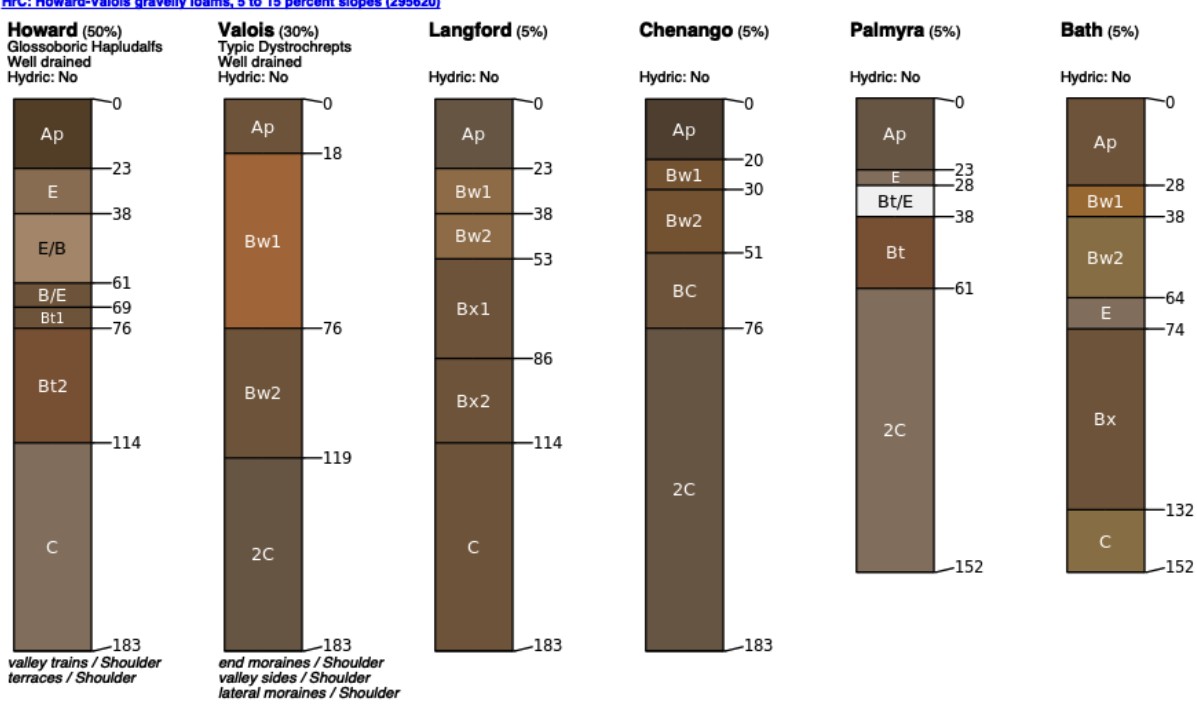

**Figure 4.** SSURGO map unit `HrC` composition at $-76°38'05"$E, $42°19'53"$N. Link to interactive map unit summary: https://casoilresource.lawr.ucdavis.edu/soil_web/list_components.php?mukey=295620

PSP predictions are closer to gNATSGO than are those of SG2, which is not surprising since PSP also uses gNATSGO as its primary information source. This product has removed some of the fine variation of gNATSGO. However the disaggregation by DSMART results in quite some discrepancies with gNATSGO. In particular, the Homer-Tully outwash valley (northeast side of map) is under-predicted by one pH unit, and the surrounding hills over-predicted by almost as much. Many of the valley 505 trains (southern side of map, running towards the Susquehanna River) are under-predicted. This is likely due to PSP's soil series predictions, which are based on estimated map unit composition and random selection of series locations within map units for DSM calibration.

### 5.3 Uncertainty

The 5%, 50%, and 95% prediction quantile maps are shown in Fig. 9 (SG2) and 10 (PSP). The "low", "representative" and 510 "high" values from gNATSGO are shown in Fig. 11. Each figure has its own stretch. gNATSGO has narrower ranges than the two DSM products and by design does not include unrealistic values. SG2 and PSP have unrealistically wide ranges at all locations. In addition, PSP shows a curious feature: fine patterning at the two extremes that is not present at the median prediction.

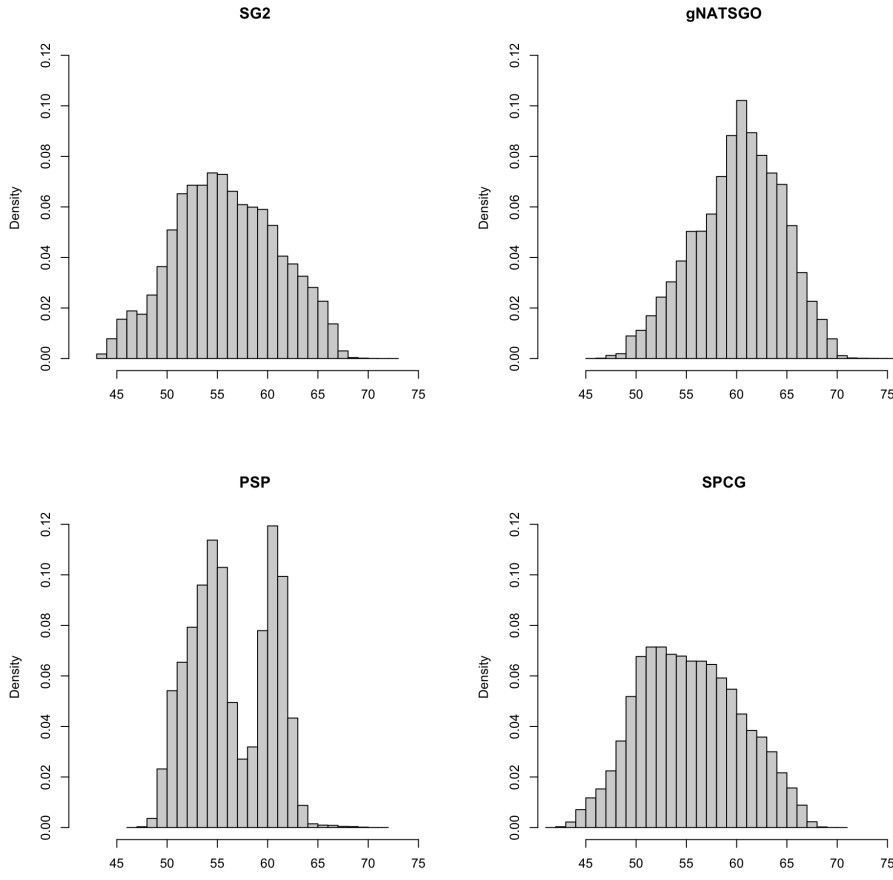

**Figure 5.** Histograms of pHx10, 0–5 cm. Note the bimodal distribution of PSP and the flatter distributions of SG2 and SPCG compared to gNATSGO.

Figure 12 shows the inter-quartile range 5–95% (IQR) for the two DSM products, along with the low-high range for gNATSGO. SG2 has a fairly consistent IQR, mostly from about 2.5 to 3.5 pH, whereas PSP has a much wider range of uncertainties, mostly from about 1.5 to 4.5 pH, and shows much more spatial pattern. PSP has the widest ranges on the steep valley sides and especially in the Seneca Army Depot at the north inter-lake area, and the lowest on the broad till plains and through valleys. These are wide ranges, and although an honest reflection of the DSM models, should give pause to map users. This suggests that the GlobalSoilMap specifications for uncertainty (Arrouays et al., 2014) are unduly pessimistic. Sources for uncertainty assessment (SG2: training points and global covariates, PSP: mapped soil series and national covariates) and the different machine learning methods lead to greatly different estimates of prediction uncertainty. The gNATSGO "low-high" range is narrower than the DSM IQR, but these are not comparable, because the expert-assigned range is not based on an estimate of a 5-95% IQR.

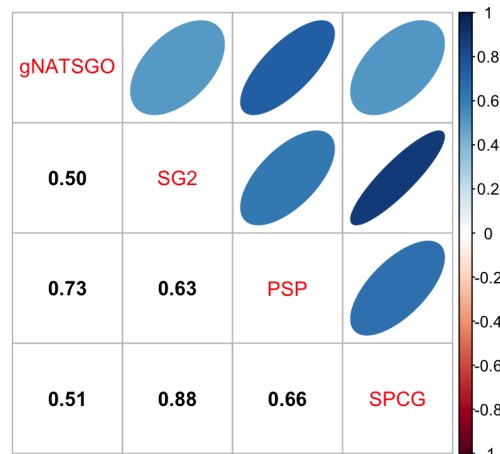

**Figure 6.** Pearson correlations between all products, pH, 0–5 cm. Strong correlations, especially between gNATSGO vs. PSP, and SG2 vs. SPCG.

Figure 13 shows the differences between the IQR of the DSM products and the low-high range from gNATSGO. Both DSM products almost everywhere have substantially wider ranges than gNATSGO, however the pattern of differences is not similar. For example, the difference with SG250 is much larger in the north of the study area, whereas PSP has the larger differences in the southern hills.

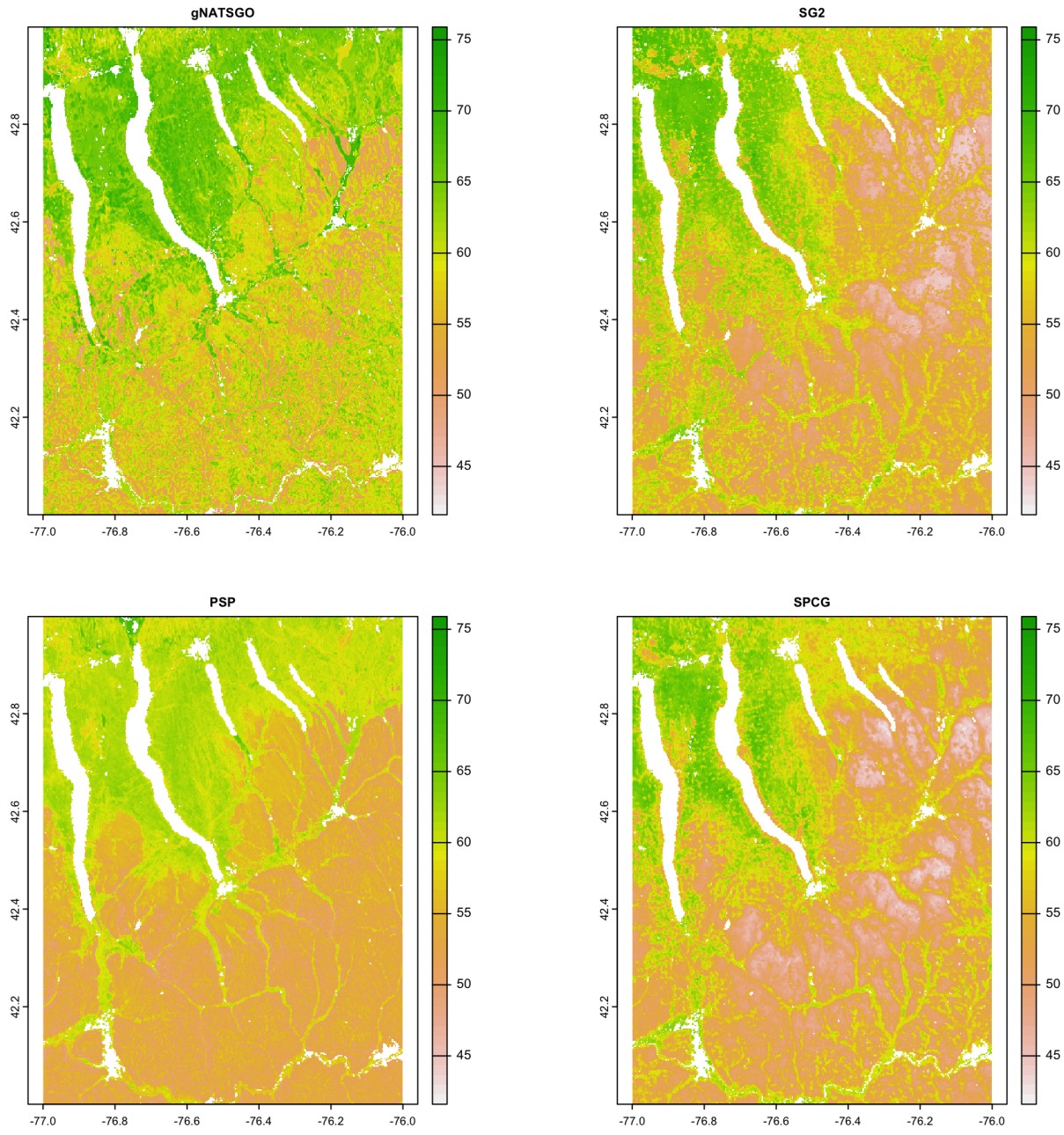

**Figure 7.** Topsoil (0–5 cm) pHx10, according to gNATSGO and DSM products. See text for discussion.

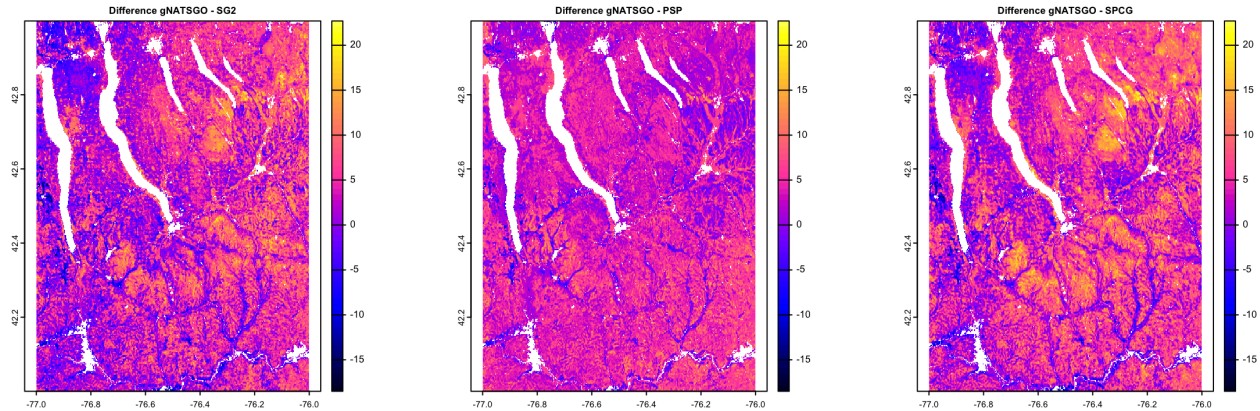

**Figure 8.** Difference between gNATSGO and DSM products, pHx10, 0–5 cm. See text for discussion.

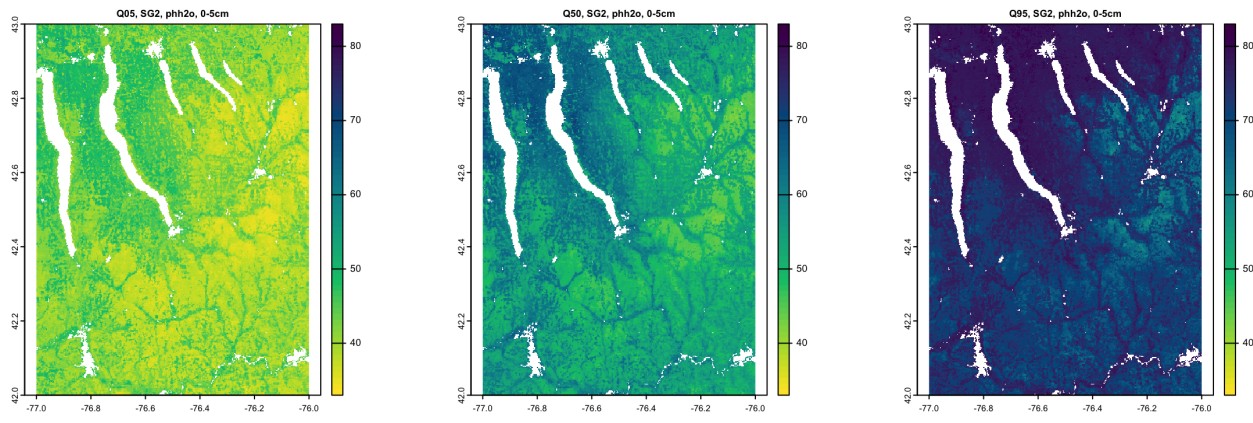

**Figure 9.** Quantiles of the prediction, SG2, pHx10, 0–5 cm. Note the unrealistically wide range at all locations and consistent patterning among quantiles.

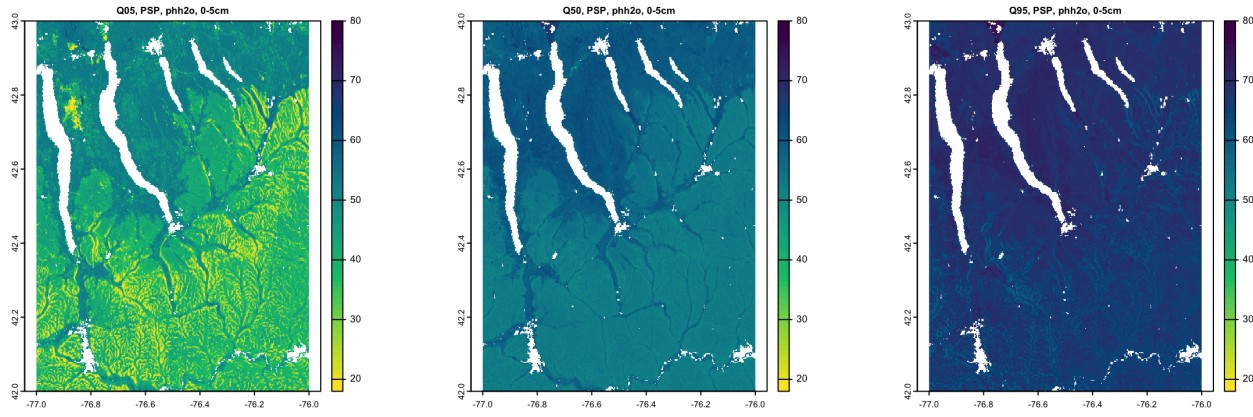

**Figure 10.** Quantiles of the prediction, PSP, pHx10, 0–5 cm. Note the unrealistically wide range at all locations and the fine patterning at the two extremes.

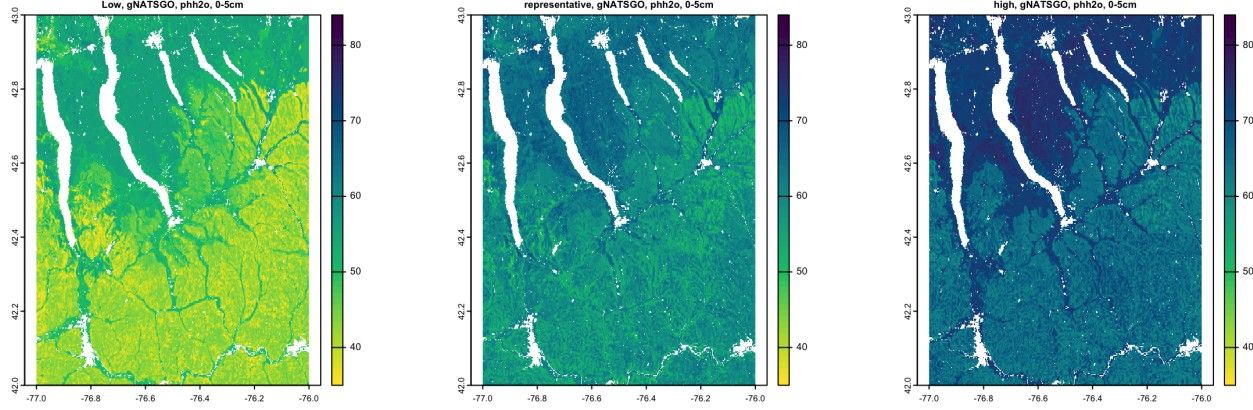

**Figure 11.** Low, representative, high values from gNATSGO, pHx10, 0–5 cm

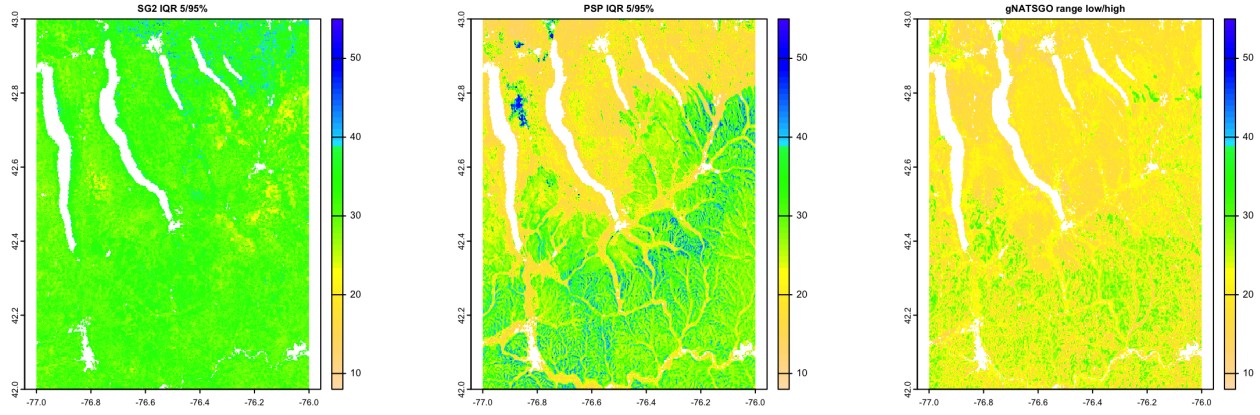

**Figure 12.** Inter-quantile ranges 0.05–0.95, pHx10, 0–5 cm. SG2 IQR is fairly consistent from about 2.5 to 3.5 pH. PSP IQR has a wider range and more spatial patterning. gNATSGO "low-high" range is narrower.

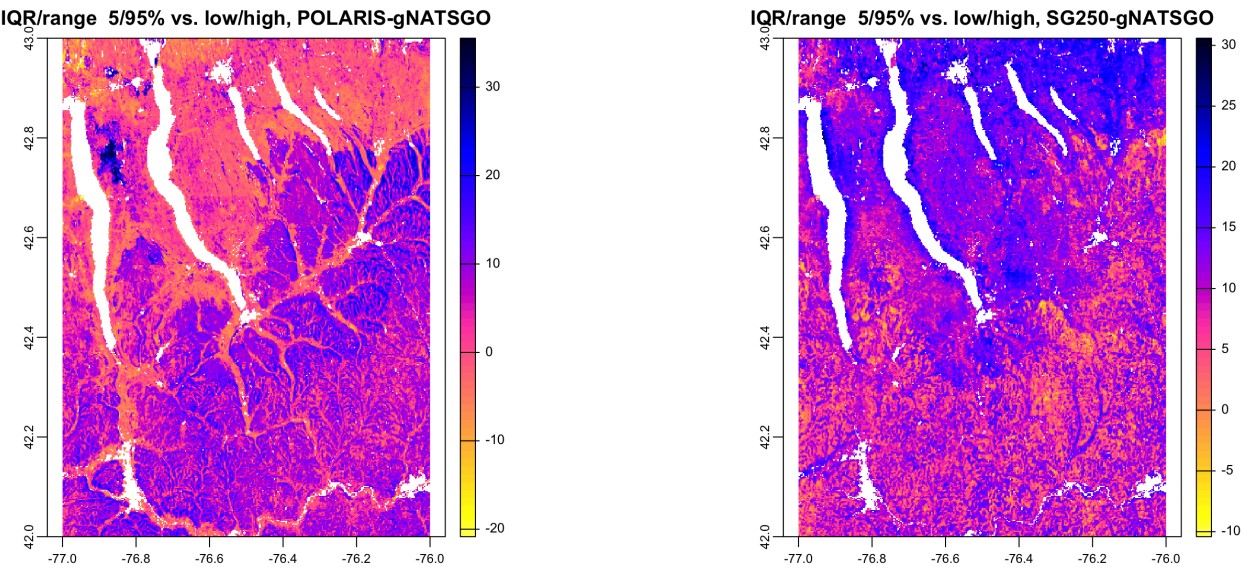

**Figure 13.** IQR/range 5/95% vs. low/high, POLARIS-gNATSGO (left), SG250-gNATSGO (right) , pHx10, 0–5 cm. Each figure has its own stretch

## 5.4 Local spatial autocorrelation

The local variograms and their fitted exponential models are shown in Fig. 14. Table 2 shows their statistics.

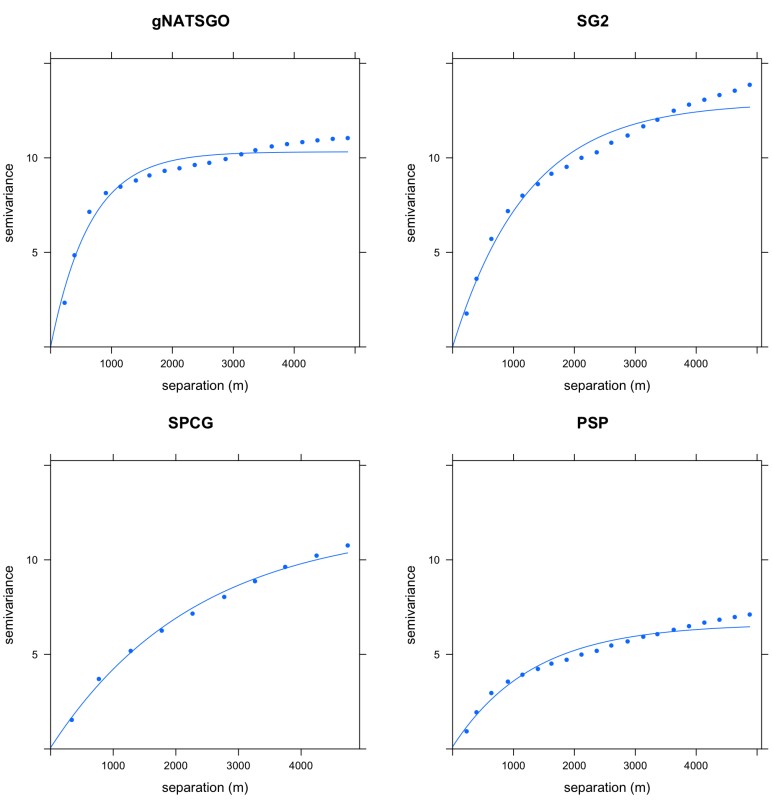

**Figure 14.** Fitted variograms, pH 0–5 cm. Semivariance units $(\mathrm{pH}x10)^2$. Note the shorter range of gNATSGO and low sill of PSP.

| Product | Effective range | Structural Sill | Proportional Nugget |
|---|---|---|---|
| gNATSGO | 1938.00 | 10.32 | 0.00 |
| SG2 | 3699.00 | 12.93 | 0.00 |
| SPCG | 6924.00 | 11.81 | 0.01 |
| PSP | 3918.00 | 6.50 | 0.02 |

**Table 2.** Fitted variogram parameters, pH 0–5 cm. Effective range in m; structural sill in $(\mathrm{pH}x10)^2$, proportional nugget on $[0\dots1]$

gNATSGO has the shortest effective range. This indicates fine-scale structure at 250 m resolution, which is of the same order as the minimum legible delineation (MLD) as a grid cell (see Introduction). The mappers who defined the boundaries between soil classes (and thus representative property values) were able to divide the landscape at this high spatial frequency, if appropriate to the soil pattern. The DSM products have longer ranges, likely due to the longer-range spatial continuity in many

| PSM_products | V_measure | Homogeneity | Completeness |
|---|---|---|---|
| gNATSGO vs. SG2 | 0.0128 | 0.0143 | 0.0116 |
| gNATSGO vs. SPCG | 0.0258 | 0.0275 | 0.0243 |
| gNATSGO vs. PSP | 0.084 | 0.0897 | 0.079 |
| SPCG vs. SG2 | 0.3342 | 0.3495 | 0.3201 |

**Table 3.** V-measure statistics, pHx10 0–5 cm

of the covariates. PSP has a longer range and lower sill than the gNATSGO from which it is derived, due to the harmonization
inherent in the DSMART algorithm. It has the highest proportional nugget, due to DSMART randomly assigning pixels within
a gNATSGO map unit to its constituents, so that neighbouring pixels may be contrasting at the shortest separation. The very
low proportional nuggets of the other products are due to the coarse resolution.

## 5.5 Classification

Figure 15 shows the topsoil pH classified into eight histogram-equalized classes in a 0.2 x 0.2° sub-tile. Class limits are
540 approximately 5.01, 5.14, 5.27, 5.40, 5.54, 5.71, and 6.02 pH, with the extreme values of 4.52 and 6.96 pH. The maps show
obvious spatial differences in class distribution. gNATSGO shows more areas in the highest pH class than the DSM products,
which is consistent with the results from continuous property maps. The pattern of gNATSGO is the coarsest, because the
classified values come from minimum-area polygons, whereas the DSM products predict per-grid cell. PSP shows the finest
spatial pattern because of its disaggregation algorithm that randomly divides gNATSGO polygons according to component
proportion. If these components are in different pH classes, there will be fine-scale pattern within the original polygon. This is
clearly the case in the large gNATSGO polygon in the northeast portion of the map (Connecticut Hill). In this example SPCG
shows large homogeneous areas of the lowest pH class, covering the highest hills, whereas SG2 presents a more nuanced view.

## 5.6 V-measure

Table 3 shows the statistics from several V-measure comparisons, based on the histogram-equalized class maps. Only SG2
and SPCG have somewhat comparable patterns. gNATSGO is considerably different from the DSM products because of its
derivation from minimum-area polygons.

Figure 16 shows the inhomogeneity and incompleteness of the SG2 pH class map (the second map for the V-measure), with
respect to the gNATSGO pH class map (the reference map). These values are the inverse of the composite values of Table 3:
the very low values in the table correspond to high values in the figure. In the homogeneity map, the blue polygons are the most
555 homogeneous areas of the SG2 map, i.e., where an SG2 polygon has the most homogeneous set of gNATSGO classified values
and thus comes closest to the reference. In the completeness map, the blue polygons are the most complete areas of the SG2
map, i.e., where the gNATSGO reference map has the most homogeneous set of SG2 classified values. The two maps have no
areas with similar patterns.

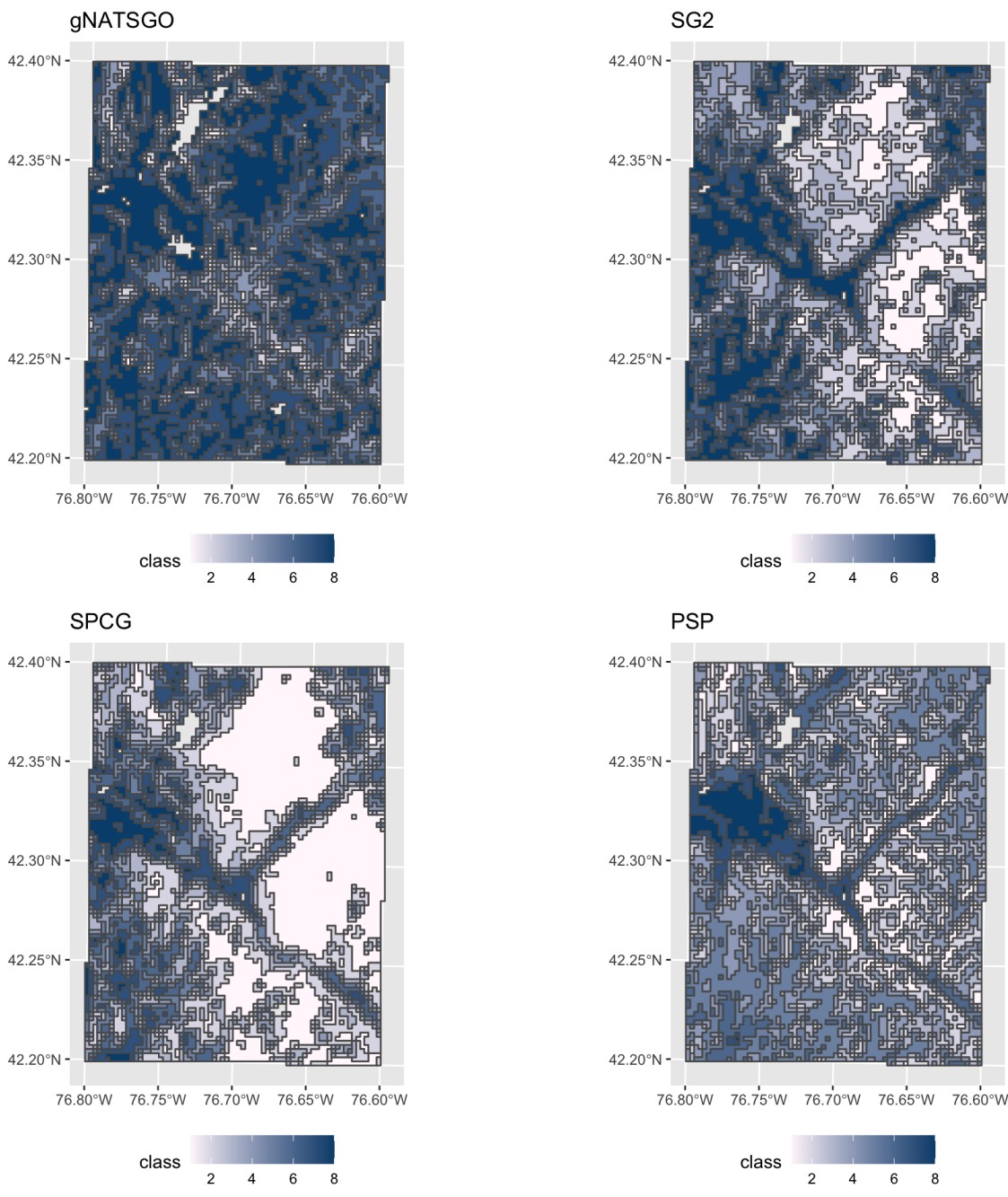

**Figure 15.** pH classes, 0–5 cm, central NY, detail. Most areas of gNATSGO are in higher pH classes. PSP has the finest spatial pattern due to the DSMART disaggregation algorithm.

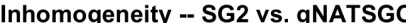

**Inhomogeneity -- SG2 vs. gNATSGO**     **Incompleteness -- SG2 vs. gNATSGO**

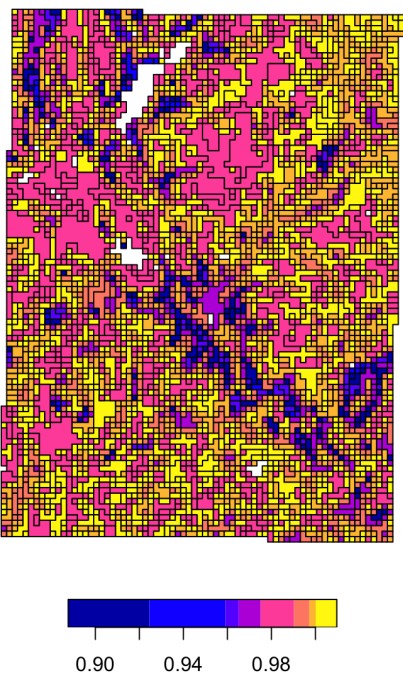
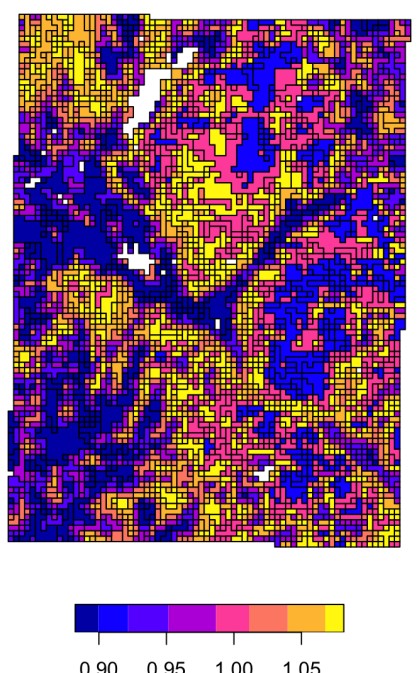

**Figure 16.** Homogeneity (left) and completeness (right) measures of the SG2 pH class map, with respect to the reference gNATSGO pH class map, 0–5 cm. Values are the inhomogeneity of each zone (left) and incompleteness of each region (right)

A contrasting result is shown in Figure 17, which compares the SG2 pH class map with respect to the SPCG map. These maps were made with similar methods and at the same resolution. The inhomogeneity and incompleteness are much lower than for the previous map, showing that the pattern of these two classified maps are fairly similar.

### 5.7   Landscape metrics

Table 4 shows the statistics from the landscape metrics calculations. The mean fractal dimensions are almost identical. There is quite some range of aggregations, with SPCG most aggregated, i.e., least complex. PSP has the most complex landscape shape, due to its fine-scale disaggregation of gSSURGO polygons. The Shannon diversity indices are highest for SG2, indicating the most even areal division into classes. This may be an artefact of the histogram equalization.

Table 5 shows the Jensen-Shannon distance beween co-occurence vectors of the four products. The co-occurence patterns of SG2 is quite similar to that of the other DSM products, whereas gNATSGO is quite different than PSP and SPCG and somewhat different than SG2. This shows that, given this histogram equalization and for the selected property and depth interval, none of the DSM products well-match the pattern from traditional soil survey.

**Inhomogeneity -- SPCG vs. SG2**

**Incompleteness -- SPCG vs. SG2**

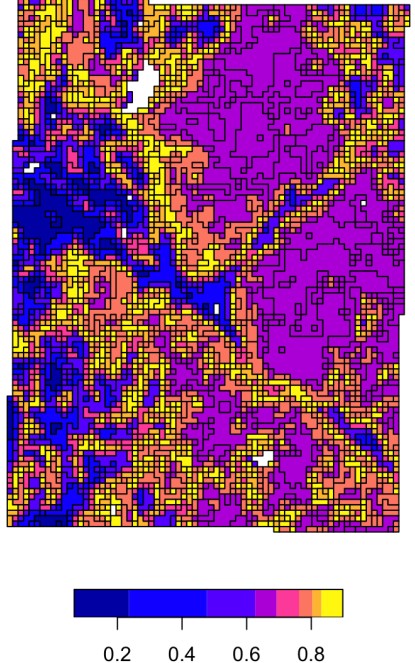
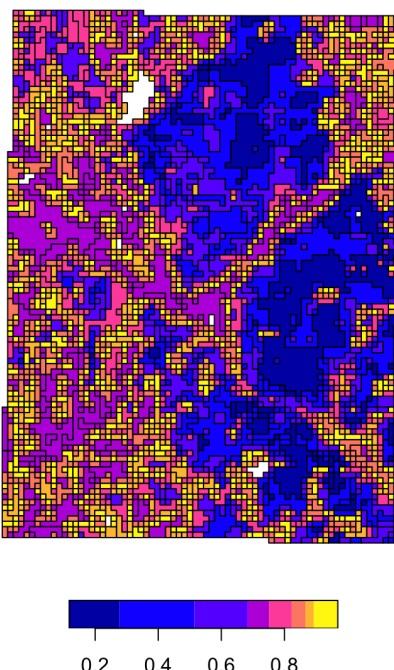

**Figure 17.** Homogeneity (left) and completeness (right) measures of the SG2 pH class map, with respect to the SPCG pH class map, 0–5 cm. Values are the inhomogeneity of each zone (left) and incompleteness of each region (right)

| product | ai | frac_mn | lsi | shdi | shei |
|---------|------|---------|--------|-------|-------|
| gNATSGO | 48.188 | 1.034 | 22.602 | 1.666 | 0.801 |
| SG2 | 50.659 | 1.034 | 21.768 | 2.06 | 0.991 |
| SPCG | 58.483 | 1.041 | 18.557 | 1.887 | 0.907 |
| PSP | 47.025 | 1.04 | 23.232 | 1.898 | 0.913 |

**Table 4.** Landscape metrics statistics, pH 0–5 cm. `frac_mn`: Mean Fractal Dimension; `lsi`: Landscape Shape Index; `shdi`: Shannon Diversity; `shei`: Shannon Evenness; `ai`: Aggregation Index

| | gNATSGO | SG2 | SPCG | PSP |
|---|---------|-------|-------|-------|
| gNATSGO | 0.000 | 0.149 | 0.281 | 0.261 |
| SG2 | 0.149 | 0.000 | 0.067 | 0.087 |
| SPCG | 0.281 | 0.067 | 0.000 | 0.111 |
| PSP | 0.261 | 0.087 | 0.111 | 0.000 |

**Table 5.** Jensen-Shannon distance beween co-occurence vectors

## 6 Local spatial patterns

The interest here is to see how well DSM methods at relatively fine resolution reproduce known relations at the local geomorphic level, e.g., hillslopes, transects across valleys with multiple terrace levels, and within farms. It has been claimed that DSM at 30 m resolution is sufficient for management of, or even within, individual farm fields. PSP is the only DSM product which predicts at this resolution.

We examine this first qualitatively, i.e., by visual inspection, and then quantitatively, mostly following the methods of the regional assessment.

### 6.1 Qualitative assessment

Here we use silt concentration, as it reveals stronger qualitative discrepancies than pH in this test area. Figure 18 shows the silt concentration of the 0–5 cm layer for (top) the gridded SSURGO overlain on the original polygons from which it was derived, and (bottom) the disaggregated PSP grid cells in a hilly landscape near Caroline, NY.

The gSSURGO product follows the SSURGO lines exactly. Some of the sharp boundary lines do correspond with abrupt transitions on the ground, for example where the steep hillsides are buried by fan alluvium. But others are not, for example on the hilltops. These differences are because the predicted silt concentrations are taken from the official series descriptions. PSP follows the map unit lines fairly well, but is much finer-grained; each 30 m pixel is separately predicted. This results in some smoothing of the abrupt boundary lines from gSSURGO on the hilltops. However within some SSURGO map units PSP predicts quite some differences in topsoil silt concentration. These are map units with contrasting components, which PSP attempts to disaggregate according to their correlation with covariates. For the most part these do not seem to be related to terrain or land use.

For example, Fig. 19 shows detail of the Holly-Papakting map unit within this PSP window. This map unit has two contrasting soils in similar proportions: a mineral alluvial soil (Holly series) and an organic soil (Papakting series); the second has much lower silt concentration.

It is difficult to see the reason for the pattern within this map unit. PSP has placed the component series in their proper proportions but not according to any apparent landscape feature or covariate.

Another example from this same area is shown in Supplementary Information §5.

### 6.2 Quantitative assessment

To see the fine differences at this high resolution, we consider a $0.15 \times 0.15°$ subtile with lower-right corner $-76.30°\mathrm{E}, 42.45°\mathrm{N}$ and evaluate pH, as in the regional assessment (§5).

Table 6 shows the statistical differences between gSSURGO (reference) and the DSM products, along with the predictions of pH. Figure 20 shows the pairwise Pearson correlations between the maps. These results are comparable to those for the full tile at regional resolution: both SG2 and PSP under-predict pH by about 0.35–0.45 pH. Correlations are fairly strong between PSP and gSSURGO, and between SG2 and PSP, but weak between SG2 and gSSURGO.

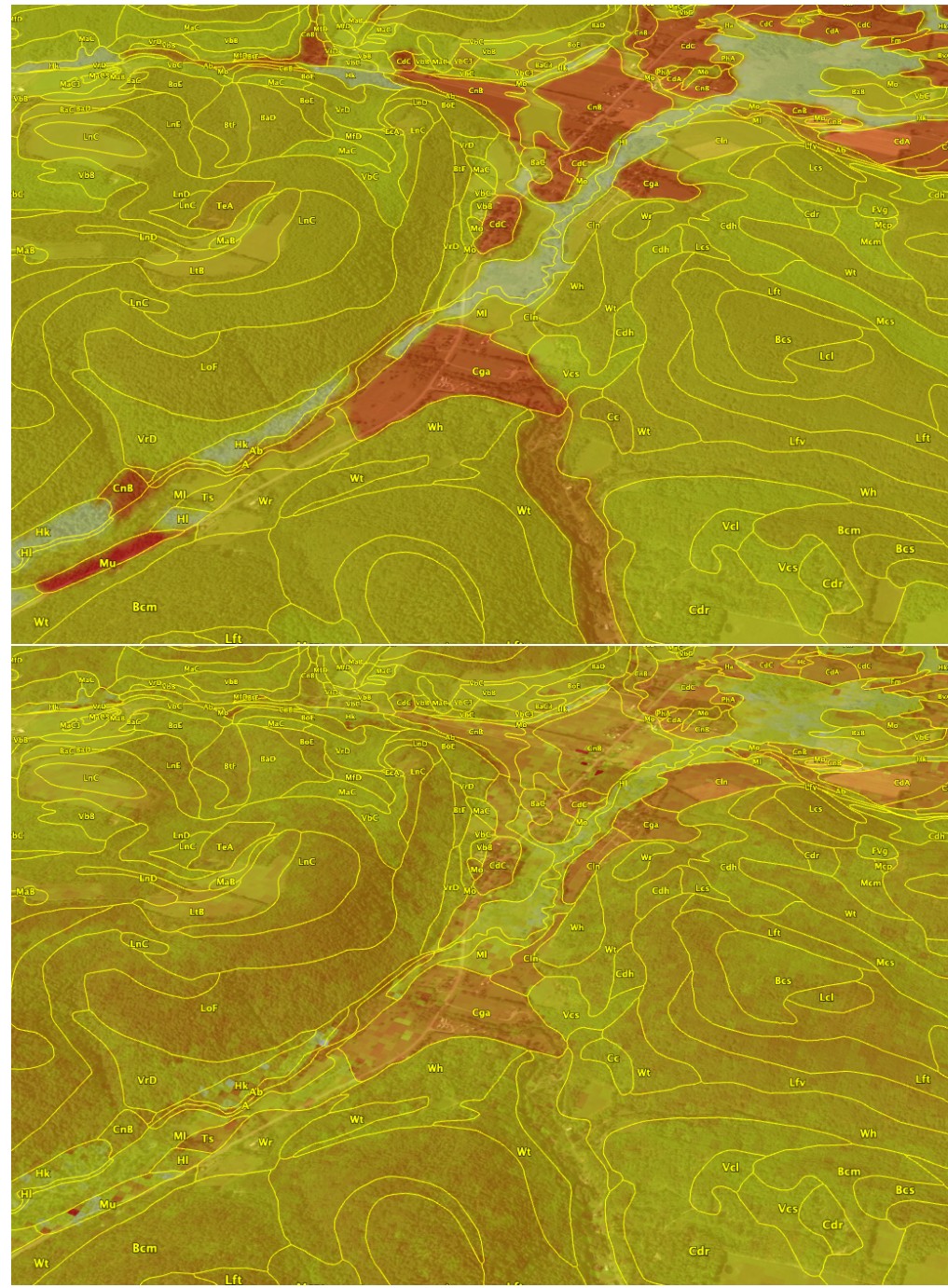

**Figure 18.** Ground overlay from gSSURGO (top) and PSP (bottom), silt % 0–5 cm, with SSURGO polygons from SoilWeb. Centre of image $-76°16'25"$ E, $42°22'53"$ N; view azimuth $247°$. Red colours are low silt, in this window alluvial fans (the C* map units). Pale grey colours are organic soils (the Hk, Hl map units). Light colours are high-silt surface soils (the L*, V*, B*, M* map units), from thin glacial till developed on shale and mudstone bedrock. gNATSGO polygons have only one value, PSP disaggregates these, hence the pixelated pattern and somewhat smoothed boundaries.

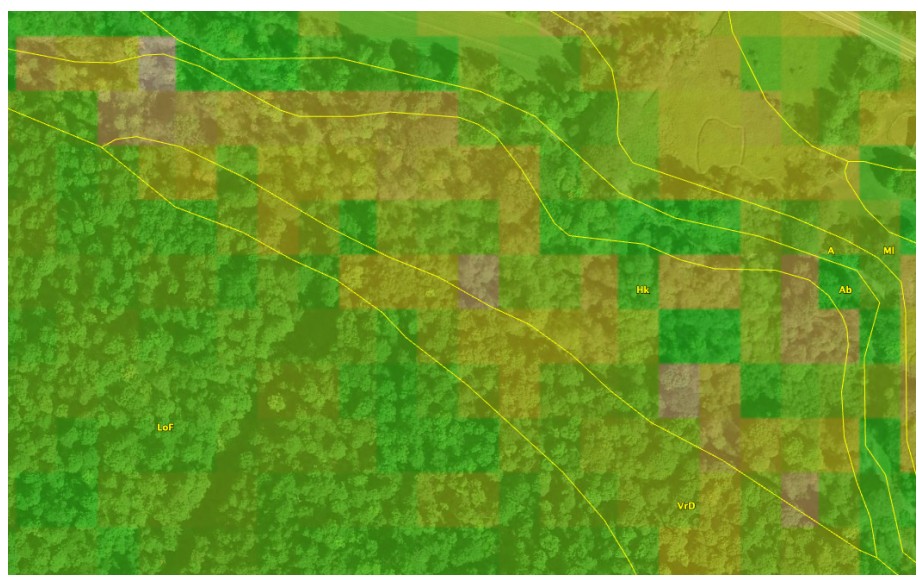

**Figure 19.** Ground overlay from PSP in the Holly-Papakting map unit, silt % 0–5 cm. Centre of image $-76°16'03"\,E, 42°22'30"\,N$. Disaggregation appears to be random and not related to covariates.

| PSM_product | MD | RMSD | RMSD.Adjusted |
|---|---|---|---|
| SG2 | 4.436 | 6.758 | 5.097 |
| PSP | 3.462 | 5.625 | 4.433 |

**Table 6.** Statistical differences between gSSURGO and DSM products, pHx10, 0–5 cm. Centre of map $-76°30'30"\,E, 42°52'30"\,N$

Figure 21 shows gSSURGO (reference) along with the predictions of pH by the PSP products. Figure 22 shows these as difference maps. Clearly, gSSURGO has overall higher values than the other two products, and despite the fine resolution, has

in general large areas of identical values. The differentiation between map units follows sharp boundaries even within a single landscape (e.g., the plateau towards the S of the map), and this is likely an artefact of relying on the representative profiles in the official series descriptions for property values. PSP has a finer pattern, due to disaggregation, and shows a smoother local pattern, without the sharp boundaries between map units within a landscape. PSP shows large areas of low pH. SG2 does not follow well the landscape lines, especially the sharp boundaries between uplands and valleys, and predicts very low pH ($\approx 4.5$)

on the plateau. It is difficult to recognize local landscape units in this global product.

### 6.2.1   Class maps

Figure 23 shows the topsoil pH classified into eight histogram-equalized classes. Class limits in this area are approximately 5.30, 5.44, 5.55, 5.61, 5.74, 5.89, and 6.15 pH, with the extreme values of 4.44 and 7.00 pH. SG2 clearly is less detailed than

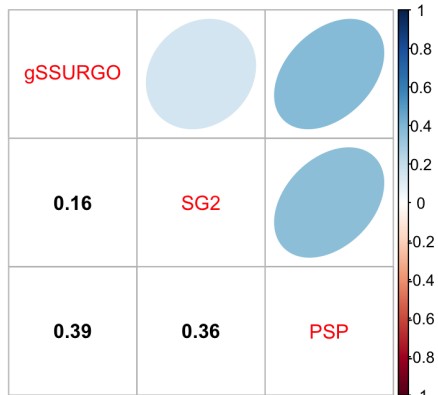

**Figure 20.** Pearson correlations between local products, pH, 0–5 cm. These moderate but weak for gSSURGO vs. SG2

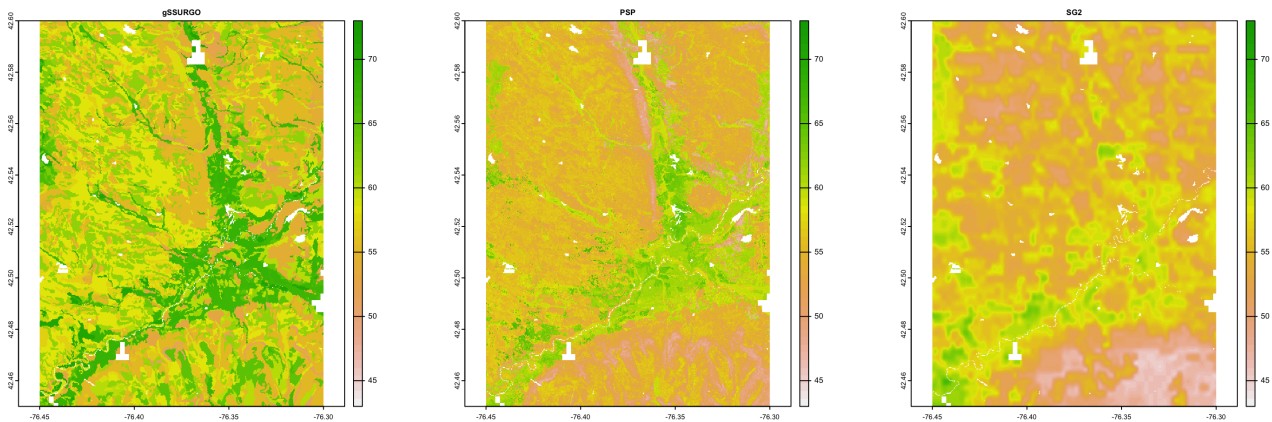

**Figure 21.** Topsoil (0–5 cm) pHx10, according to gSSURGO and DSM products. See text for discussion.

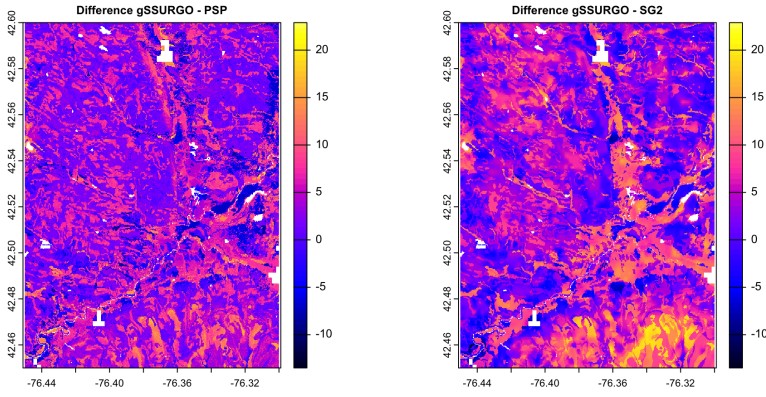

**Figure 22.** Difference between gSSURGO and DSM products, pHx10, 0–5 cm. See text for discussion.

the other two products. PSP shows a fine pattern, not closely related to the fine pattern of gSSURGO. As previously noted,
gSSURGO is consistently about one pH class higher than the other products.

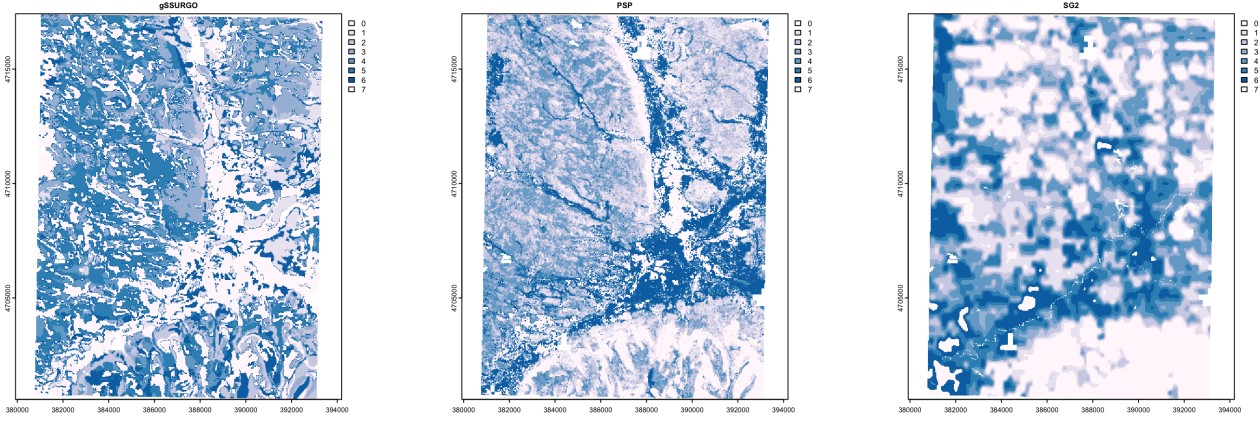

**Figure 23.** pH classes, 0–5 cm. Coordinates are UTM 18N meters.

### 6.2.2 Local spatial autocorrelation

The local variograms and their fitted exponential models are shown in Fig. 24. Table 7 shows their statistics. gSSURGO has
the shortest effective range and highest sill. PSP has a longer range and low sill, due to the harmonization from DSMART that
removes some of the overall variability. SG2 has no nugget variance, a low sill, and long range, consistent with its regional
scale.

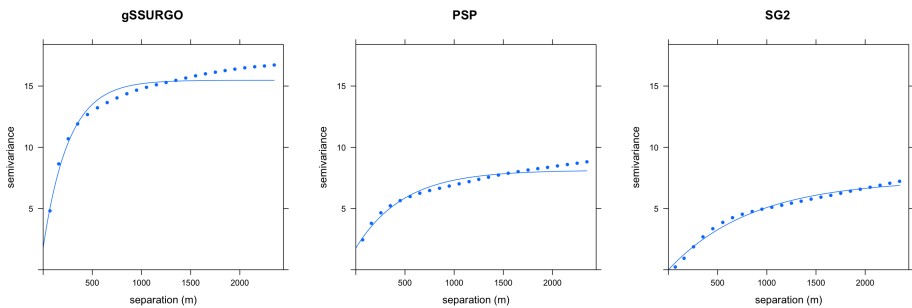

**Figure 24.** Fitted variograms, pH 0–5 cm. Semivariance units $(\text{pH}x10)^2$. Note the short range of gSSURGO and low sill of SG2 and PSP.

| Product | Effective range | Structural Sill | Proportional Nugget |
|---|---|---|---|
| gSSURGO | 774.00 | 13.67 | 0.12 |
| SG2 | 2550.00 | 7.34 | 0.00 |
| PSP | 1455.00 | 6.36 | 0.22 |

**Table 7.** Fitted variogram parameters, pH 0–5 cm. Effective range in m; structural sill in $(\text{pH}x10)^2$, proportional nugget on $[0\dots1]$

| product | ai | frac_mn | lsi | shdi | shei |
|---|---|---|---|---|---|
| gSSURGO | 73.658 | 1.049 | 71.395 | 1.845 | 0.887 |
| SG2 | 87.647 | 1.106 | 34.978 | 1.941 | 0.934 |
| PSP | 56.376 | 1.045 | 116.476 | 2.006 | 0.965 |

**Table 8.** Landscape metrics statistics (local), pH 0–5 cm. `frac_mn`: Mean Fractal Dimension; `lsi`: Landscape Shape Index; `shdi`: Shannon Diversity; `shei`: Shannon Evenness; `ai`: Aggregation Index

| | gSSURGO | SG2 | PSP |
|---|---|---|---|
| gSSURGO | 0.000 | 0.218 | 0.168 |
| SG2 | 0.218 | 0.000 | 0.112 |
| PSP | 0.168 | 0.112 | 0.000 |

**Table 9.** Jensen-Shannon distance beween co-occurence vectors (local)

### 6.2.3 Landscape metrics

Table 8 shows the statistics from the landscape metrics calculations. The mean fractal dimensions are almost identical. SG2 is much more aggregated, i.e., least complex, than gSSURGO or PSP. PSP has a higher landscape shape and Shannon diversity than the other products. Table 9 shows the Jensen-Shannon distance beween co-occurence vectors of the four products. The co-occurence patterns of SG2 is somewhat similar to that PSP but quite different from gSSURGO.

## 7 Conclusions

The presented methods are well-able to expose differences in maps produced by different DSM mapping methods, and between these and traditional soil survey. There are also well-documented differences between maps based produced by traditional survey methods. For example, Bie and Beckett (1973) compared four independent surveys of a 19 km$^2$ area in Cyprus, and found that the maps differed considerably in their map unit purity and their proportions of interclass and intraclass variability. Thus the use of the NRCS products as reference should be seen as a basis for comparison with DSM products, not as the "truth". However, it is the best available representation the of soil landscape at the given design scale and legend.

Our methods for comparing maps have two limitations due to the decision that they be applicable, using the supplied computer code, to any area within the USA. The first is the use of histogram equalization for the class maps which are then evaluated for the class pattern. For specific areas and properties it would be preferable to use established class limits relevant for land use, for example limits from soil survey interpretation tables. The second is the choice of the exponential model for automatic variogram fitting, as well as the somewhat arbitrary choice of empirical variogram cutoff and bin width. For each area, property and depth interval variograms could be computed and fit according to the analysts' prior knowledge.

A variety of metrics to compare DSM products among themselves and to the reference map are proposed in this paper. This raises several questions about their utility and possible redundancy. Since we only consider one example case here, our conclusions are tentative. Comparing the results here with those in the companion case studies report, we know that these are context-dependent, and no general conclusions can be drawn. All the metrics provide useful information based on different summaries of the maps, so none are redundant.

A first question is which metrics best reflect the visual differences in patterns, e.g., for the regional patterns of Fig. 7. For the continuous maps, the whole-map histograms (Fig. 5) reveal whether the feature-space distribution of the property known from gNATSGO has been distorted by the DSM method. <<<<< HEAD In the example case PSP produced a strongly bimodal distribution, so that its map shows few values near pH 5.8. ======= In the example case PSP produced a strongly bimodal distribution, so its map showed few values near pH 5.8. >>>>> 56de475b489470b570c995fd75098582716df8e9 Much of the patterning in the strongly acid soil region is homogenized towards lower values, and there is a sharper boundary between the strongly and moderately acid areas. By contrast, both SG2 and SPCG reduced the peak modal value pH 6 and made more predictions towards the two tails of the univariate distribution. This can be seen in the resulting maps by more areas with the colours towards the two ends of the colour ramp. The whole-map variograms (Fig. 14) reveal the longer-range spatial continuity of the DSM products, compared to gNATSGO. This can be seen in the maps as less fine detail and larger areas with similar values.

For the classified maps (e.g., Fig. 15), the large discrepancies between them is due to the slicing from histogram equalization. Each landscape metric (Table 4) reveals a different aspect of the maps. For example, the aggregation index `ai` shows that SPCG contains much larger one-class areas, on average, than the other products, and this is clear in the figure. Consistent with this, the landscape shape index `lsi` shows that SPCG has a simpler overall shape.

A second question is which metrics best discriminate the different DSM products. The whole-map histograms and variograms clearly show which products are more similar. In the example case, SG2 and SPCG are quite close and PSP is substantially different by both these metrics. The Jensen-Shannon distance beween co-occurence vectors (Table 5) clearly shows dissimilarity in the adjacency patterns of classes. In the example case again SG2 and SPCG are quite close, but here PSP is not too different. The landscape metrics were inconsistent in this case.

It is clear from the "best case" example presented in this paper that different DSM methods, with different training points, covariates, and algorithms, can produce quite different predictive soil maps. Thus comparing maps with point-wise evaluation from (almost always biased) field observations gives an incomplete picture of how the different methods represent the soil landscape, which is after all what dictates how the soil is used and managed.

The main findings from the example case are:

1. Although the regional products (250 m resolution) are well-correlated, the DSM products are biased, under-predicting topsoil pH by about 0.38–0.48 pH units. They also differ substantially, with a RMSD adjusted for bias on the order of 0.31–0.48 pH. This is based on representative pH values of the mapped STU, not on measured values.

2. The DSM products differ substantially among themselves and with the reference product in their local spatial pattern, as revealed by empirical variograms. gNATSGO has a short effective range, but this is smoothed to a range 2 to 3.5 times as long by DSM.

3. Classification by histogram equalization reveals major differences in the spatial patterns of the produced class maps, as evaluated both by visual inspection and landscape metrics.

4. Despite using USA-specific covariates (parent material, drainage classes) derived from gNATSGO and covariates limited in geographic scope to the USA, the predictive map made by SPCG is not substantially different from that made by SG2, likely due to the similar modelling method.

5. The estimates of uncertainty provided by SG2 and PSP are substantially different, both in width of the uncertainty interval (much narrower in SG2) and in spatial pattern. This could be in part because SG2 is a global model, whereas PSP is based on local soil surveys and covariates restricted to one tile. The confidence intervals seem unrealistically wide compared to the expert-derived high-low value range provided by gNATSGO.

6. At the local level (30 m resolution) the disaggregation provided by PSP does not appear to correspond to landscape positions associated with STU components. PSP obscures the fine-scale details of the local spatial pattern, and SG is substantially more general, due to its resolution.

These results will differ in different soil geographic regions, for different soil properties, and for different depth intervals, as shown in the companion Case Studies report.

Why are the results from these DSM examples so poor? Why do they not better approximate traditional surveys? We present some possible reasons:

1. The dominant DSM methods do not explicitly consider spatial continuity or pattern. Experiments have been started with convolutional neural networks and other methods with varying window sizes of covariates.

2. Environmental covariates to represent past soil-forming conditions (the "time" factor) are only available since the satellite remote sensing age, very short in terms of soil formation.

3. Environmental covariates to represent soil parent material (e.g., surficial geology) are not available globally, and even for the USA the proxy of using parent material derived from SSURGO in SPCG was not of sufficient precision to improve the predictive models.

4. Point observations were mostly placed by the soil surveyor at "typical" or "representative" locations in order to characterize map units, and do not capture the full range of variability along toposequences.

5. Poor georeference of legacy point observations, many from the pre-GPS era, leads to poor correlation with environmental covariates, hence to poor models, hence to much noise in the DSM product, which can obscure patterns.

6. Traditional soil survey uses is also a predictive activity. The surveyors uses as "covariates" (i.e., non-soil environmental information related to soil geography) what can be inferred from airphotos, as well as direct landscape observation (terrain, vegetation, land use etc.). These give a more detailed and nuanced view of than possible at the resolutions used in practical DSM at regional scale, i.e., 100 m or coarser.

Despite the discrepancies between DSM products and field survey, DSM can be a valuable tool for soil survey. Because of the expense and difficulty of field survey, in practice DSM is likely to be the most used method of making or updating soil maps in areas with no or poorly-resources soil survey organizations. For unsurveyed areas DSM can provide a useful pre-map for planning sampling and field survey, thereby optimizing scarce resources for field work. It has the advantage of being reproducible and objective, given a set of training points, relevant environmental covariates, and a machine-learning method. Many of its problematic results are due to a set of training points, often with imprecise georeference, that do not properly occupy the covariate feature space and, and to the lack of covariates to represent some aspects of pedogenesis over time.

In the USA (our study area) and in other countries with active soil survey programmes, DSM will be an important but not dominant tool in the overall survey. Soil survey as practiced by the NRCS uses methods from DSM, applied statistical modeling, and numerical ecology, along with an active and focused field programme. For example, supervised classification of terrain derivatives and satellite imagery has been successfully used to check internal consistency of map unit concepts and assist with the placement of delineations. The aim is to blend the most applicable tools from traditional field survey and applied statistical methods, supported by pedologic theory and regional land use considerations.

Of course, soil survey must be based on a proper examination of the soil itself. There is no substitute for actually examining the soil and landscape, for either traditional soil survey, or as a reliable basis for DSM.

*Code availability.* Source code as R Markdown documents are available at https://github.com/ncss-tech/compare-psm. These can be used to (1) import all products to compare, as well as some others not considered in this study; (2) create ground overlays and corresponding KML files for display in Google Earth; (3) compare SG2 and PSP for $1 \times 1°$ tiles; (4) compare SG2 with SPCG and gNATSGO for any rectangular tile; (5) compute landscape metrics and compare them between products for any subtile of these; (6) evaluate the success of PSP
in disaggregating at 30 m resolution.

*Author contributions.* DGR conceptualized the approach, did most of the writing, wrote the R Markdown documents and performed the example case study. LP provided DSM expertise and detailed knowledge of SG2. DB and ZL provided USA-specific expertise, in particular about the NRCS and its products and services. All authors collaborated on the motivation, methods and conclusions.

*Competing interests.* There are no competing interests.

*Acknowledgements.* The contribution of Zamir Libohova was mostly accomplished during his tenure at USDA-NRCS-National Soil Survey Center, 100 Centennial Mall North, Room 152, Lincoln, NE 68508-3866 USA.

His contribution was partly supported by the LE STUDIUM Loire Valley Institute for Advanced Studies through its LE STUDIUM Research Consortium Programme.

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
