# Peer review of "How well does Digital Soil Mapping represent soil geography? An investigation from the USA"

_SOIL, 2021_

## Author Response (AR1)

SOIL-2021-80
Author's Response to comments

These are separated by RC and CC.

Other changes introduced in preparing the revision:

1. Supplementary Information \S5 (comparison of IQR ranges of SG2 and PSP with the gNATSGO high-low ranges) moved into the main text. This short section complements the previous discussion of this topic.

2. Added references to webpages describing gNATSGO and gSSURGO.

3. Added relevant references published or brought to our attention since the original submission:

Kupfer, J. A.: Landscape ecology and biogeography: Rethinking landscape metrics in a post-FRAGSTATS landscape, Prog. Phys. Geogr., 36, 400-420, https://doi.org/10.1177/0309133312439594, 2012.

Meyer, H. and Pebesma, E.: Machine learning-based global maps of ecological variables and the challenge of assessing them, Nat Commun, 13, 2208, https://doi.org/10.1038/s41467-022-29838-9, 2022.

Meyer, H., Reudenbach, C., Hengl, T., Katurji, M., and Nauss, T.: Improving performance of spatio-temporal machine learning models using forward feature selection and target-oriented validation, Environmental Modelling & Software, 101, 1-9, https://doi.org/10.1016/j.envsoft.2017.12.001, 2018.

Pindral, S., Kot, R., Hulisz, P., and Charzyński, P.: Landscape metrics as a tool for analysis of urban pedodiversity, Land Degradation & Development, 31, 2281-2294, https://doi.org/10.1002/ldr.3601, 2020.

4. Took the opportunity to polish the language and be as specific as possible in our meaning.
* * *
SOIL-2021-80 Reply to RC1

We appreciate the reviewer's careful summary and appreciation of our objective.

1. Reply to summary and general comment:

1.1 "The examples given in the paper do not fully demonstrate that the proposed metrics are necessary, relevant and sufficient to provide a comprehensive evaluation of the different DSM products that can help an unexperienced end-user in choosing between the different available DSM products he/she can get in a given local territory."

We could not come to any conclusion on this -- indeed as the reviewer points out, "No clear and convincing hierarchy across the three DSM products is revealed by the analysis of the result."

The reviewer presents the following three questions to be addressed in the Discussion:

Q: "Which metric best account for the visual differences of soil patterns?"

Q: "Which metrics are redundant to each others?"

Q: "Which metrics best discriminate the different DSM products?"

In the Conclusions we have directly addressed these.

1.2. "The authors used a fairly unusual way to present their data (section 2) i.e., instead of presenting separately each DSM and soil survey products, they chose to deliver the information progressively and "in parallel" along a set of sections that are not always straightforward for an external reader. Furthermore, this induces some redundancies and contradictions."

The "redundancies and contradictions" are dealt with in the response to detailed comments (see below). Other than that, we are not clear on why the presentation is "not always straightforward". There is a lot to explain about the products, so that their differences can be (somewhat) explained by their characteristics.

1.3 "The study areas selected as examples for regional and local spatial patterns are, if not presented (the local ), not presented with the necessary details (regional) for allowing the interpretation of the results".

These are discussed in detail in the companion Case Studies document, as the first of the four case studies. We omitted them from this paper in order to focus on the methodology and to keep the approach generic. Our aim is to show how an assessment is performed, not primarily to assess this specific case study. However we agree this would aid understanding, so we have repeated the context information from the Case Study document in the main paper \S4, as requested.

1.4 "The presentation of the soil data considered in this paper (actually in sections 2 [Data sources] and 4 [Example area and soil property]) needs to be deeply reworked to improve the understanding by an external reader not familiar with US context."

As explained above, this presentation is already in the Case Studies, but since many readers of this paper will not go to the Case Studies, we have repeated the soil data and context explanations here in \S2, and also refer to this specific case in the Conclusions.

2. Reply to comments along the text: (1) Those of form/consistency have been dealt with by appropriate edits; (2) Those of substance are discussed now.

2.1 "Title and line 1: Why do the authors rename "predictive soil mapping" what is currently known as "Digital Soil Mapping"? Indeed, why? The first author has preferred the idea of computer-generated maps as "predictive", and has followed the terminology of Scull (1993). However as pointed out in the Community Comment from the University of Sydney, all maps are in some sense predictive, and DSM is by far the most-used term. The revised paper, including the title, will be adjusted accordingly.

2.2 "Line 114: "data sources" is only the first part of the section (before 2.1.) isn't it? If yes you should replace "data source" by "soil data" and add a subsection "data source" immediately after."

Indeed this is not an appropriate heading. We have changed it to "Products compared"; this then segues into the next primary section "Evaluation methods".

2.3 "Line 138: Contrary to what is suggested here "Polaris soil properties" is not further

systematically replaced by "PSP". A lot of "POLARIS" remains in the text and in the figure. This should be corrected."

We are sorry for our lack of care, and have made the text consistent. We have used PSP for the soil properties product of the POLARIS project, which also includes the soil class predictions of the original POLARIS (since updated).

2.4 "Line 152: gSSURGO has not been presented before"

The explanation of gSSURGO and its distinction from gNATSGO has been added to the paragraph beginning at L122 of the Discussion paper.

2.5 "why mixing in a same section "environmental Covariates" and "geographic scope"? The relation between them is weak. The statements that deal with the latter (Lines 191–193 and 195) would be better located in the next "mapping methods" section (see my next comment)"

We combined these because the geographic scope of the covariates used differs among the DSM products. We wanted to emphasize that SPCG only uses covariates covering CONUS, and that PSP uses only covariates within each of its tiles, while SG2 uses global covariates.

2.6 "perhaps more comfortable for the reader to know first the set of covariates used by SG2 and also, as I understood, by the two other DSM products. Then cite the specific covariates that have been added to PSP and SPCG."

2.7 "Lines 197–205 : this paragraph is largely redundant with lines 151–161."

2.8 "Lines 218–224 : insert here the geographical scope and the number of samples used for training the models."

2.9 "Line 231: I understand here that gNATSGO is a generalization of gSSURGO. However it was stated before ( line 125) that "gNATSGO is a composite of [....] SSURGO and [....] STATGEO and [....] RSS. Please, explain more clearly what are the differences between gNATSGO and gSSURGO."

Indeed there is confusion here. This has been explained in the paragraph starting at L122 (as explained in comment 2.4).

2.10 "Lines 255–257: Even before looking at your further results, we could expect that soil survey products and DSM products do not converge toward similar uncertainties assessments. At best, we could expect that the level of uncertainties mapped by these two products could be ranked similarly, independently from their absolute values. You should select a metric for representing this."

We used the IQR (5–95%) as per GlobalSoilMap specifications to characterize SG2 and PSP; these are compatible. The inclusion of the low–high estimate range from gNATSGO is purely informative, in the revision we have made it clear that this is based on expert opinion backed up (in some cases) by pedon data, but is not meant as specific points on a probability distribution.

2.11 "Line 272: "RMSD adjusted for MD". Not clear. A mathematical formula would clarify."

We think text alone is sufficient, if we rephrase this as "RMSD adjusted for MD, i.e., the RMSD after subtracting the bias, i.e., the MD, from each prediction."

2.12 "Line 301: "region" is qualitative. How can we calculate the variance of a qualitative

variable?"

The regions are not qualitative, they are geographic extents of the histogram-equalized class map (note that 'zones' is used for the same concept on the reference map).

2.13 "Lines 359-375: The sections "Regional patterns" and "local patterns" are redundant with the introduction of section 3 (lines 259-265). This should be re-organized."

2.14 "Lines 377-349 (section 3.6.1. "visual method") Does this section refer only to "local patterns" (section 3.6.1.)? I don't think so since you provided further visual comparisons for both regional and local patterns. This should be clarified. Furthermore, this section looks redundant with section 3.1. ("qualitative methods")"

This is within \S.6 "Local patterns" so refers to that. S\3.1 "Qualitative methods" only refers to side-to-side visual comparisons, not overlays on Google Earth.

2.15 "Figure 1: This study area looks different from the ones considered further (Figures 4 to 12). This could explain why there is a so great discrepancy between the visual inspection and the quantitative results obtained further. Similar problem occurs with figure 13 (what is this study area?) . Please give the length and width of the rectangle for a better appreciation of the scale."

This is now Figure 3 (geographic context is now Fig. 1 and 2). And indeed it was incorrect! The background was supposed to be SG2 but was incorrectly SSURGO. This has been corrected. A scale bar has been added to show the scale; also the 250m pixels of SG2 clearly show the scale.

2.16 "Lines 393-402 (section 4): It would be useful to have more information about the study area that is finally selected as example of regional spatial pattern comparisons (size of the rectangle, scale of the soil survey product gNATSGO at this location, average size of polygons, pedology, landscape drivers of soil variability etc...). All these data would be very useful for interpreting the results. This comment applies also to the description of the study area selected further as example of local spatial patterns comparisons. I did not find any information on this area."

This is explained above (1.3). Indeed this information greatly helps the interpretation of the relative success of the DSM products.

2.17 "Line 412: provide the significance of the sizes of the circles in figure 3. Indicate what you mean by "well-correlated" (threshold?)

Line 464: There is an apparent contradiction with the concluding statement here "overall the agreement is fairly good" and what is written just before (line 459):

"gNATSGO is considerably different from all other products""

Indeed that is not a correct conclusion, which we see when the V-measures are computed. Removed.

2.18 "Lines 539-541: I suppose you should replace "...and covariates limited in geographic scope to the USA" by "...and input soil data limited in geographic scope to the USA". To my opinion, this is the most surprising result of the paper. Have you any explanation ? to my opinion, the similarity of machine learning algorithms cannot be a convincing explanation."

It is so that the covariate coverage was limited to the USA, this was not made clear. The

covariates are (mostly) the same but in this case the model does not try to use covariates outside the USA -- although that wouldn't make a difference anyway, because no points for model training would be located there. We have clarified the statement as suggested.

We were also quite surprised by this result. We've thought some more about this and have added some text with tentative explanations. We do still think that the machine learning algorithm is part of the explanation.

2.19 "Line 543-544. you cannot conclude that because the specifications for defining these confidence intervals are different between the DSM and the soil survey products. Furthermore, there is not any ground truth to identify what is the most realistic CI."

Correct, there is no ground truth for the CI. However the CI is applied point-wise (actually, grid-cell-wise)

2.10 "Lines 556-560. I disagree with your diagnosis on PSP. Figure 14 clearly shows that PSP does not bring more knowledge on soil variations than the initial soil survey product from which it was derived. Furthermore, you cannot say that PSP can be useful for unsurveyed areas since PSP requires a soil map as input"

We think the reviewer means that what we say in general about PSM in the indicated lines does not apply to the PSM method of (what we will, in the revision, call) DSM.

(1) We can not find where we claimed that PSP (or any DSM method) brings more knowledge on soil variations than the initial soil survey product; certainly we don't say that in the indicated lines. We do say that some covariates used in DSM show variation not considered by the original mappers. An example is \gamma-ray surveys. Therefore the output of DSM should be compared with the field survey to see if it might reveal otherwise unrecognized patterns.

Figure 14 shows that PSP does disaggregate polygons (although in this case incorrectly, we think). It's true that the knowledge for the disaggregation comes from the initial soil survey product, so in that sense the reviewer is correct, there is no more knowledge.

(2) Indeed PSP can be used to extrapolate: it takes the constructed model and applies it to the covariate stack in the extrapolation area. It has been calibrated on a map of soil series (and from these, their soil properties) in a mapped area, but once the model is built it can be applied anywhere -- a set of soil series will be predicted, and from that, soil properties. Of course this may not be wise.
* * *
SOIL-2021-80 Reply to RC2

We thank Prof. Miller for his careful review, including some fundamental questions to be addressed, as well as confusions to be cleared up.

1. Reply to summary and general comment:

1.1 "The spatial analysis techniques presented in this manuscript introduce important evaluations that need to be considered when comparing soil map products. However, the manuscript would benefit from a more thoughtful consideration of the meaning of the evaluations chosen. The metrics selected offer a variety of calculations, but it seems possible that they may be reflecting some of the same differences between the maps. Even after attempting to mute the effect of resolution, some patterns in the results remain. For example, some smoothing should probably be expected from models that – at their core –

rely on regression fitting. So there appears to be some opportunity here to go beyond just reporting differences in the metrics. The metrics are intended to measure map characteristics other than relative smoothness. Can all the differences detected by the metrics be attributed to relative smoothness, and if not, what other map characteristics might these metrics be picking up on? I advocate for this because if others are to be convinced to apply these evaluations elsewhere, they will likely want to have some sense of how the results should be interpreted and/or what should be learned from them."

Thank you for this key observation. We admit to being unclear on why certain metrics show certain similarities and differences, in some cases contrary to our expectations. We have added several long paragraphs to the Conclusions to discuss this. We have also reviewed the description of each metric in \S3, where we have explained why the metric is used and what it might show. Note that we had already done this, especially for \S3.4.2 "Landscape metrics", but this has been expanded and clarified.

1.2 "The spatial analysis of map properties promoted in the present manuscript reminds me of the books "Soil and Landscape Analysis" by Hole and Campbell (1985) and "Pattern of the Soil Cover" by Fridland (1977). Although dated in their focus on analyzing polygons, these books suggested several ways additional information could be evaluated from the spatial patterns in the map. Then again, the methods proposed here essentially go back to a polygon type of analysis since they require the map to be classified."

Agreed. Those books were in fact the inspiration for this study. In a revision have added some background to pattern analysis in soil survey, including these books, as part of the paragraph at L80ff.: "It has long been recognized that the soil cover forms patterns at various scales (Fridland 1974, Hole and Campbell 1985), and the traditional soil mapper attempts to find those patterns that are expressed at the map design scale." We prefer the more accessible 1974 Geoderma paper from Fridland to his book, which in any case is referred to in the paper.

1.3 "The way in which the mapping methods for the different products are described is sometimes unsupported and potentially not fully relevant to the spatial analysis applied. The distinctions between machine-learning models (categorized as PSM) and traditional soil mapping seem questionable. If the authors want to keep these assertions, then citations or better explanations for why they think they are true need to be added. A building concern in the description of traditional soil survey is a potential failure to recognize what the two approaches being compared have in common, which is important for understanding the differences in the respective products."

We think that there is a major difference between DSM, which proceeds from observations and covariates to a map, with traditional survey, which depends on expert interpretation of the soil-landscape, supported by purposive observations to support/reject/modify the surveyor's mental model. We have brought out this contrast in the first paragraph of the paper.

These differences are indeed relevant to the analysis. Traditional methods stratify the landscape into polygons based on expert judgement and landscape analysis, whereas DSM methods do not stratify as such, they predict per-grid cell, using covariates that are meant to represent soil forming factors.

As for the mapping methods of the different products, we don't see how our descriptions are "unsupported". We cite the original papers for all three DSM products, and the description of the NRCS products is directly from their descriptions by the NRCS, and by the two co-authors who have worked in the NRCS soil survey.

It is difficult to more fully answer this comment without reference to specific texts that the reviewer thinks are deficient. However, we will review our explanations for clarity.

1.4 "To improve the manuscript, I would encourage the authors to consider how the strategies of the different mapping approaches may be connected to the results of the spatial evaluation metrics applied. As an example, if we were to recognize traditional soil survey as a predictive map product, then the covariates would largely be what the mapper could see in the available airphoto. Although the airphoto bases used for traditional soil survey leave a lot to be desired compared to modern covariates, one can see more detail of landform shapes than with a 100m digital terrain data set. Although resolution is already partially alluded to in the text, this kind of context may help sort through what may be driving some of the differences detected with the various map evaluation metrics."

This is a good point. We did not discuss traditional approaches in any detail, as our aim was to compare their results with DSM products, and then within these. The sentence about traditional methods mention above does suggest what the surveyor is looking at. We have expanded on this in the Conclusion, when discussing why DSM may give disappointing results, modifying the reviewer's point somewhat:

"Traditional soil survey uses is also a predictive activity. The surveyors uses as ``covariates'' (i.e., non-soil environmental information related to soil geography) what can be inferred from airphotos, as well as direct landscape observation (terrain, vegetation, land use etc.). These give a more detailed and nuanced view of than possible at the resolutions used in practical DSM at regional scale, i.e., 100~m or coarser."

As for airphotos, these are 3D views that can be used to identify landscape units that can then be used as aggregating units for displaying soil properties. Of course, this approach is rather simplistic when it comes to soil property patterns as we all know, the distribution of certain properties may follow other patterns rather than just a landscape or landforms. Therefore in detailed survey the polygons delimited on airphotos may be separated based on field examination of the soil.

1.5 "While I offer considerable critique on the characterization of traditional soil survey and push for some consideration about what the selected metrics really describe about the maps being evaluated, I applaud the spatial analysis approach to evaluating different map products. Thinking about how to evaluate maps beyond the prediction of points is an important contribution to the advancement of soil mapping as a science."

We appreciate the reviewer's encouragement, and hope to have dealt with the above main issue, and the general points and details below. Indeed "how to evaluate maps beyond the prediction of points" was our main motivation for this work.

General Comments

2. "Please consider being more consistent with terminology and abbreviations. For example, traditional soil survey versus conventional soil survey and map scale versus design scale. They appear to be used interchangeably, but the change in term makes the reader wonder if there is a difference being implied and if so, what that difference may be. Regarding terminology, I disagree with several of the terms used in the present manuscript but attempt to use the terms most frequently used by the authors in the present manuscript to facilitate communication in this review of the manuscript."

Indeed we were sloppy here.

(1) In the revision we have used "traditional" survey throughout, and explained what we mean

by that. Immediately after the first mention of DSM have added "This [DSM] is in contrast to what we here call ``traditional'' soil survey, in which the soil surveyor develops a mental model of the soil geography (Hudson 1992) by interpreting the landscape with the aid of airphotos, purposive transects, and detailed profile descriptions at locations thought to represent the central concepts of the soil classes present in the study area" and here citing the Soil Survey Manual. We then replaced the "conventional soil maps" with "polygon maps made by traditional soil survey methods".

(2) "I disagree with several of the terms used in the present manuscript" we would like to know which and why.

3. "Information presented in sections 1 through 2.3 sometimes circles back on itself. Consider reorganizing to avoid repeating some information. I think this would also help readers understand and compare the processes by which these maps are made along with the resulting map characteristics."

Indeed there is some redundancy here, which we have edited out.

Section 1 has no information on the specific products, only a discussion of the overall problem and approach. Then in Section 2 we broke down the differences between products into (1) nature of the maps to compare, (2) the primary data on which they are based, (3) the environmental covariates used in DSM. We intend to remove the heading of (1), as it follows directly from the section heading. (2) can also be folded into the earlier description of the NRCS products. This section has been tightened and some information moved to \S2.4 "Mapping Methods". This should make the differences between products easier to understand.

4."Most figure captions are more like titles. Please elaborate in the figure captions to guide the reader in what to look for in the figures."

Our practice has been to point out the most important features of a figure in the text, at the point the figure is referenced. In some cases this text would be too long to fit into a caption. However have expand most figure texts to make them independently understandable. For some figures the discussion in the text can not easily be summarized for a figure caption. In these cases we will add "See text for explanation".

Specific Comments

L53-55 – "The acknowledgment of the Scull et al (2003) paper using the term 'predictive soil mapping' is appreciated. However, I'm concerned that the use of this term could lead to confusion over what differentiates traditional soil survey from the new approaches that utilize computational algorithms to produce soil maps. The potential issue here is the perception that traditional soil survey is somehow not predictive. Of course, it is not possible to observe soil everywhere, which requires some level of spatial prediction for any kind of soil map with exhaustive coverage."

Yes, this was pointed out by RC1 and some CC, and is correct. "PSM" was replaced by "DSM" throughout. Scull is still cited and the previous term "PSM" acknowledged.

L62 – "How fair is it to say that machine-learning models can be implemented with fewer locations visited when the machine-learning models presented for comparison in this manuscript rely on a database of observations collected by the activity of traditional soil mapping? In statistical theory, it makes sense that some optimized sampling design should be able to capture the needed information. But has this been the experience of soil mapping?"

Here we are not saying that the maps presented here depend on fewer points. In this general introduction about the attractions of DSM, one of them is indeed that with clever sampling in covariate space -- if the relation with covariates is strong (as stated in this sentence) -- fewer points can be used than if the standards for sampling density at various map scales are followed. However, as the reviewer points out, this has not, to our knowledge, been studied as a direct comparison between traditional and digital soil mapping. This is partly because traditional maps are quite difficult to evaluate quantitatively, unlike DSM which at least can do some cross-validation.

So we have softened the statement to:

"Further, it may be that fewer locations can be visited in order to develop reliable models, as compared to traditional survey techniques. If the relation with covariates is strong, and  locations representative of the entire covariate feature space are included in the training set, it may be possible to map large areas from relatively few field observations."

We also have referred here (as in L190) to the "homosoil" concept: identical environmental conditions (as represented by covariates) should result in the same soils.

L64 – "Please explain how PSM has a greater ability to map inaccessible areas than traditional methods."

DSM can extrapolate based on similarity in covariates, again the "homosoil" concept, with the caution (as we write) "if the available training data cover the covariate space of the inaccessible area." This is essential the same idea as the "Konzeptbodenkarten" (conceptual soil maps) long used in German practice to map an area from covariates (in the past, polygon maps of various themes such as geology and relief) prior to fieldwork. As an early and thorough example see Grenzius, R.: Konzeptbodenkarten für den städlischen Raum, Z. Pflanzen. Bodenk., 156, 209-212, https://doi.org/10.1002/jpln.19931560304, 1993, where maps of topography, historical land use, biotypes, flooding, climate, etc. are used to make a concept map. This could be called a "pre-map". But in the case of concept maps there is not yet soil information, this will come from an appropriately stratified survey. In the case of DSM extrapolation, the soil information comes from the training area and is extrapolated based on the covariates and machine-learning model.

We could have added an explanation here, but this introduction is not meant to be a full exposition of DSM.

L73 – "The assumption that mapping scale drives resolution is largely a concept held over from paper maps. In many disciplines (e.g., geology) we are seeing newer generations of maps adding detail without changing the extent of the map. Adding those details (making the resolution finer) has become functionally possible because the producer and the user can 'zoom' in and out of the map in a GIS. This technological development renders the question of mapping scale nearly mute. However, the remaining question is if there is sufficient information to support the finer resolution."

We disagree. Mapping scale and resolution are concepts from two different concepts of space (polygon and grid) but are related by the spatial detail of information. Scale is not extent, it is the ratio of the physical map to real world, e.g., 1:50 000.  The paper map can of course be reproduced with a larger piece of paper but this does not add information. So the design scale of a polygon product, based on the concept of minimum legible delineation (Forbes, T. R., Rossiter, D., and Van Wambeke, A.: Guidelines for evaluating the adequacy of soil resource inventories, Cornell University Department of Agronomy, Ithaca, NY, 1982) is related to the density of information. Tom Hengl (among others) has related the two

concepts, see Hengl, T.: Finding the right pixel size, 32, 1283–1298, https://doi.org/10.1016/j.cageo.2005.11.008, 2006.

We disagree that the ability to zoom in renders the question of scale moot ("having little or no practical relevance", often because in law the situation that gave rise to a proceeding is no longer relevant). If the information density is not sufficient there is nothing more to see. We discuss the selection of a proper pixel size in the context of DSM in the SOIL paper presenting SoilGrids v2.0 (Poggio, L., de Sousa, L. M., Batjes, N. H., Heuvelink, G. B. M., Kempen, B., Ribeiro, E., and Rossiter, D.: SoilGrids 2.0: producing soil information for the globe with quantified spatial uncertainty, 7, 217–240, https://doi.org/10.5194/soil-7-217-2021, 2021)

L80 – "This paragraph appears to present a non sequitur. The argument is made that point evaluations of PSM do not consider the spatial pattern of predictions. However, it is not clear to me how the subsequent information presented about traditional soil survey methods shows how that approach does more to evaluate spatial patterns of predictions."

This observation is correct. We were contrasting with the traditional approach, and implicitly suggesting that map users would compare the pattern of polygons (map unit delineations) with the landscape, and also that the pattern of polygons is immediately visible to the map user, as opposed to the pixelated results of DSM.  We have re-written this paragraph as:

"A more serious issue is that point evaluations of DSM products do not consider the spatial pattern of predictions. By contrast, traditional soil surveys produce polygon maps of relatively homogeneous pedons, with the boundary lines placed at inflection points of maximum change between soil bodies (Lagacherie 1996). These maps explicitly show the surveyor's interpretation of the soil landscape as developed from a mental model of the soil-forming processes, and which when viewed as a whole show the pattern of the soil cover. It has long been recognized that the soil cover forms patterns at various scales (Fridland 1974, Hole and Campbell 1985}, so that the traditional soil mapper attempts to find those patterns expressed at the map design scale. Since DSM predictions are on a grid cell basis, there is no concept of relatively homogeneous natural soil bodies nor inflection points between them. However, it might be expected that if the valus of the DSM covariates representing the soil-forming factors also cluster in a similar pattern to the soil cover, the DSM predictions would also cluster and approximate map units from traditional survey, despite being predicted separately per grid cell. The question is thus to what degree DSM products represent the actual soil landscape spatial pattern and, more importantly, the underlying pedogenetic and geomorphic processes."

We think this explains the difference between traditional and DSM methods of pattern formation.

Another point  is that the selection of training points is based on spatial patterns trying to be mapped, i.e., landform units. Soil surveyors know that this is circular reasoning, but the landscape model, rather than covariate space, is the vehicle for selecting training points. In theory they should produce similar results if the points used in DSM were indeed training points for landforms units approach of SSURGO. PSP tries to find virtual training points within SSURGO map units to match with covariates, but apparently did not well succeed in the presented example (and in the examples of the companion ISRIC Report).

L83-88 – "This section appears to be building an assumption that traditional soil survey does not include any kind of model that uses input data to make predictions. Although the authors point out real shortcomings in the "paradigm" of traditional soil survey described by Hudson (1992), they have left out how that approach uses 'mental models'. This omission obscures what machine-learning and traditional soil survey methods have in common."

See the previous comment (L80); we think the re-write brings out this point.

L91-95 – "The cartographic reasons that traditional soil survey is constrained to the levels of detail it has is a useful explanation here. However, for translating polygons delineated in USA soil survey maps, there is a more direct approach. The USA soil survey program has a strict protocol for minimum delineation areas. The USA's "Field Book for Describing and Sampling Soils" specifies minimum-size delineations of 0.6 ha for 1:12,000, 1.0 ha for 1:15,840, and 2.3 ha for 1:24,000. Now, we should note that those are minimum delineation sizes, and the mean delineation size will be larger than that. The mean delineation size can vary by landscape and by the style of the mapper."

It's clear that these are taken directly from the Forbes et al. criteria, probably because Richard Arnold was at Cornell when they were being developed, and then went on to head the NRCS Soil Survey.  We have now specified that these are in the Field Book and have cite it.

L126 – "The use of STATSGO2 is interesting here. With the possible exception of some areas where only an order 5 map has been made, STATSGO is a purposefully generalized map product that is aggregated by expert knowledge. At first, I questioned if it made sense to include STATSGO in this evaluation, but then STATSGO was not evaluated. Considering STATSGO is not evaluated in this manuscript, it does not seem relevant."

STATSGO is used to fill in areas without more detailed survey. This is explained in the gNATSGO description and in L130.

L128 –" "State" should not be capitalized."

Has been corrected, also in several other locations where this is not part of a proper noun.

L162-169 –" This is a nice, succinct description of the state of SSURGO."

Thank you.

L175-183 – "I think it may be misleading to state that SG2 does not use any information from SSURGO when SG2 uses profiles from the NRCS pedon database. Yes, SG2 is not using SSURGO itself, but they are both using a set of training points that they have in common for the USA area. This overlap in source information is even more so for the SPCG, which makes use of additional pedons that were produced from the activities of the USA Soil Survey. In the case of the RaCA dataset, it is new enough that it probably has not strongly influenced the SSURGO map. Nevertheless, the role of these training points in all the map products should be explained clearly. Specifically, I disagree with the idea that SSURGO is independent of the data points managed by the NRCS."

This was poorly-worded. We meant that SSURGO polygons were not used. Indeed SG2 uses NRCS profiles, which were also (indirectly) used to build SSURGO. We have now made it clear that SSURGO is a map product, and does not include the points.

From https://www.nrcs.usda.gov/wps/portal/nrcs/detail/soils/survey/?cid=nrcs142p2_053627: "SSURGO datasets consist of map data, tabular data, and information about how the maps and tables were created." This only applies to the polygons. So we have added "any information derived from SSURGO polygons ..."  We also have added that SPCG uses SSURGO map units to derive parent material and drainage classes.

L199 – "Add space after the period."

Done. How did that slip through the editing process?

L247–252 – "Libohova et al. (2014) explored the validity of these ranges to represent uncertainty. That evaluation seems relevant here."

Thank you for reminding us to include this reference: Libohova, Z., Wills, S., and Odgers, N. P.: Legacy data quality and uncertainty estimation for United States GlobalSoilMap products, in: GlobalSoilMap: Basis of the Global Spatial Soil Information System, Boca Raton, 63–68, 2014. We added "Libohova et al. (2014) discuss how these estimates can be derived for USA products following the GlobalSoilMap.net specifications."

L267 – "Consider a rubric here to define how the expert judgement will be evaluating the maps. This will help the reader understand the value system being applied in this evaluation and communicate a more structured approach to how the qualitative comparison will be made."

Each expert will see things differently, but there are some commonalities. We have explained a general approach ("rubric"):

"The DSM product can be evaluated at selected known points, typically from field observation of test areas: is the ``correct'' soil type or property  predicted? and if not, is the error a reasonable approximation? More interesting for our purposes are patterns in the DSM product. These can be compared to patterns used in the mental model of traditional soil survey, for example, toposequences and sequences of parent material outcrops. In both cases (points and patterns) the evaluator may be able to infer which DSM covariates would be needed to improve the map."

L283 – "Where does variability come from for any single point in these maps. Won't there be a single value for a grid cell, or is there something else being brought in here? Is this using the uncertainty ranges? In any case, please explain clearly to help the reader know the basis for the proportional nugget."

There is one value per grid cell. But when these are plotted on an empirical variogram the smooth function fit to the increasing separations in geographic space vs. these in feature space often does not intercept the semivariance axis at 0, when the separation is 0. This is attributed to many causes, e.g., lab error or variability within the grid cell (which is represented only by its centre). This is a common phenomenon in variography for soil data. L283–4 "The proportional nugget shows the inherent variability at a point, at a scale shorter than the grid spacing" is correct, we will modify the sentence to explain the possible causes of this.  Because of smoothing there is generally low or no nugget variance in DSM products. We have changed this sentence to "The proportional nugget shows the variability at the prediction point at the centre of a grid cell, at a scale shorter than the grid spacing."

L383–L388 – "This paragraph drifts into results by beginning the evaluation. Recommend keeping the description of methods separate from the results found by implementing them."

Agreed. We had this figure here to illustrate the method, but indeed these are results, and have been moved to a first subsection of the "Regional spatial patterns" results, with some more explanation that is only a preliminary evaluation.

Table 1 – "Please be consistent in abbreviations."

Has been changed to match the use of abbreviations in the text.

L408 – "Change 'distributions' to 'distribution'?"

Done.

L429 – "Add missing 's' after 'PSP'"

Done.

Figure 3 – "Consider including the r values in the boxes."

Table 2 – "Again, please be consistent in abbreviations, both for matching abbreviations used in the text and previous tables."

Has been changed to match the use of abbreviations in the text. The entire document has been checked for consistency.

L446 – "add 's' at end of 'indicate'"

Done.

Figure 4 – "The 'SoilGrids' and 'SPCG100' maps show large areas of pH lower than shown in the 'gNATSGO' in the high elevation portions of the east and south areas. Could this be a case of extrapolation in the feature space? If so, how might this be reflected in the evaluation metrics presented in the is manuscript?"

Random forests do not extrapolate beyond their training values, so this can not be the cause. In this case it is likely that the higher elevations in the Catskills (to the southeast of this study area), which have similar forest land cover, have biased these predictions. In fact, the NASIS database (incorporated into WoSIS) does have several points in these higher hills, and these show the lower pH (around 5.0) of the topsoils, even in forest, as opposed to the pH 4.5 predictions from SG2 and SPCG. A local model covering only this tile might well have results that are more consistent with gNATSGO.

L447-448 – "The authors suggest that the smoothing effect was caused by the spatial continuity of the covariates, which seems reasonable considering the resolution. It seems to me that any kind of fitted model is, almost by design, going to smooth out some patterns in the training data. Would the authors mind commenting on this possible additional factor?"

Good point. We have commented on this. Re-thinking the original comment, not all covariates are smooth (e.g., vegetation indices can vary greatly pixel-to-pixel if land use varies at the scale of the grid cell. There is more we now say about these results:

"gNATSGO has the shortest effective range. This indicates fine-scale structure at 250 m resolution, which is of the same order as the minimum legible delineation (MLD) as a grid cell (see Introduction). The mappers who defined the boundaries between soil classes (and thus representative property values) were able to divide the landscape at this high spatial frequency, if appropriate to the soil pattern. The DSM products have longer ranges, likely due to the longer-range spatial continuity in many of the covariates. PSP has a longer range and lower sill than the gNATSGO from which it is derived, due to the harmonization inherent in the DSMART algorithm. It has the highest proportional nugget, due to DSMART randomly assigning pixels within a gNATSGO map unit to its constituents, so that neighbouring pixels may be contrasting at the shortest separation. The very low proportional nuggets of the other products are due to the coarse resolution."

Figure 10 – "Did the 'SPCG100USA' semivariogram actually reach a sill?"

In an exponential variogram model the sill is never reached; by convention 95% of the fitted sill parameter is considered the effective range. The total sill is 11.82 (Table 2), at the presented cutoff the variogram reaches about 11, which is 93% of the total sill. So not quite 95% but close enough. It's clear from the nice fit to the exponential model form that the variogram is asymptotic to a sill and not increasing without bounds.

L462–463 – "These sentences are a little unclear; please consider rewording and/or expanding upon the explanation."

Yes, the homogeneity and completeness are not intuitive. They were explained in \S3.4.1 "V-measure", which we now referenced here. However the caption and explanation were in fact incorrect, we missed that the sense of these maps is reversed from that of the accompanying table.

Also while reviewing this section, we decided that it would be informative to also show the relation between SG2 and SPCG, because these are much closer in method, and in fact show much more similar patterns, which is revealed by these maps. We have discuss these differences, pointing out the reason for them:

"Figure 15 shows the inhomogeneity and incompleteness of the SG2 pH class map (the second map for the V-measure), with respect to the gNATSGO pH class map (the reference map). These values are the inverse of the composite values of Table 3: the very low values in the table correspond to high values in the figure. In the homogeneity map, the blue polygons are the most homogeneous areas of the SG2 map, i.e., where an SG2 polygon has the most homogeneous set of gNATSGO classified values and thus comes closest to the reference. In the completeness map, the blue polygons are the most complete areas of the SG2 map, i.e., where the gNATSGO reference map has the most homogeneous set of SG2 classified values. The two maps have no areas with similar patterns."

The second comparison has been similarly discussed and then contrasted with the first comparison.

In addition \S3.4.1 has been expanded and better explained.

L468–469 – "If the difference between gNATSGO and PSP is going to be called out, maybe it is worth mentioning that the difference between gNATSGO and SPCG is even more. Some discussion about why the difference between gNATSGO and PSP captured the authors' attention may also be warranted."

We were confused by this result. After re-thinking, we have added this text: "The co-occurence patterns of SG2 is quite similar to that of the other DSM products, whereas gNATSGO is quite different than PSP and SPCG and somewhat different than SG2. This shows that, given this histogram equalization and for the selected property and depth slice, none of the DSM products well-match the pattern from traditional soil survey."

L473–474 – "Just to be clear, which PSM product is being referred to here?"

PSP. This was not clear. "This" has been replaced by "PSP".

L478 – "This is the first mention of silt concentration! This switch makes for a mismatch between the methods described and the results presented."

Indeed. This section is meant to show how PSP, which claims to disaggregated gSSURGO, is not always (usually?) successful. In this study area the topsoil pH does not differ too much in local landscapes (although it does have a wide range across a 1x1 degree tile), so is not a

suitable example of what we want to show. We explain this as "Here we use silt concentration, as it reveals stronger qualitative discrepancies than pH in this test area."

L480–482 – "This content would be better suited in the figure caption."

Correct. Was so moved.
* * *
SOIL-2021-80 Reply to RC3

1. This is a timely article on a subject of importance to many involved in soil survey programs. As the metrics show, some 400 views have been made in the United States alone.

A: Thank you for this acknowledgement. Indeed we hope this stimulates further studies in this direction. And the linked code allows anyone to make our same comparisons for their area and properties of interest in the CONUS. These could be interesting papers, since the authors would be familiar with the soil geography of their areas.

2. Many soil scientists and administrators involved in soil survey see Digital Soil Mapping (DSM) as more than just a tool to aid the field soil scientists mapping soils using traditional methods. In their view DSM is the new method for mapping soils. Artificial intelligence and machine learning are tremendous tools in many medical and scientific studies, and it is logical to conclude that these methods when applied to soil survey will generate significant results, which they may. Yet, an assessment of DSM methods at this stage of development is needed that can articulate strengths, weaknesses, and opportunities.

A: Agreed. However this paper is not about a SWOT of DSM in general, rather on how to compare of DSM products, and vs. a "traditional" product. We do discuss DSM in general but that is not the main focus.

3. The assessment that is made in this paper uses visual and statistical techniques that compare the DSM methods of POLARIS, SoilGrids, and SPCG to gSSURGO (and gNATSGO), which are used as references. Many challenges to DSM have come from field soil scientists using traditional methods—that is, an understanding of soil genesis, geomorphology, and Quaternary geology of the soils being mapped combined with on-site hypothesis testing. Authors of this study have such field experience as well as backgrounds in computer-assisted soil survey studies.

A: Thank you. Authors Beaudette (NRCS) and Libohova (formerly NRCS, now ARS, and with extensive field experience during his Purdue PhD time) are the co-authors with the most expertise.

4. McBratney et al. object to comparing DSM to gNATSGO as the reference, stating that both DSM and "conventional" soil mapping have uncertainties and often a different focus (classes vs properties), why should one be used to measure the quality of the other? "Would we not reach a similar conclusion if we take maps from different soil surveyors to compared them with a DSM product? In order to do a convincing comparison, it is important to have an independently observed dataset with which to compare the various representations else we might simply realise a self-fulfilling prophecy." I predict that such a comparison will soon be made and, thus, reinforces the merit of this paper for moving the science forward.

A: Agreed. This is not the focus of our paper. However, in the Conclusion we have made a reference to a very early, but still relevant, study comparing soil surveyors: Bie, S. W. and Beckett, P. H. T.: Comparison of four independent soil surveys by air-photo interpretation, Paphos area (Cyprus), 29, 189-202, 1973.

From the Abstract to that paper: "'[T]he four interpreters used quite different strategies for mapping the same soil landscape, to produce soil maps which differed considerably in the percentage purity of their mapping units [based on 30 profiles] and the extent to which the variability of soil properties within mapping units was less than that of the landscape as a whole." So which of the four maps would be used as a "reference"?

5. Comments about retaining the familiar term "digital soil mapping" instead of the new term "predictive soil mapping" are reasonable. Overall, the paper generated a lot of discussion and thought and will be a good contribution to the literature. It may prompt some in the DSM community to reflect on whether DSM is over-sold, at least in some cases. The paper also makes a contribution by explaining terms to the non-specialists, such as SSURGO, STATSGO, WoSIS, NASIS, SPCG, etc.

A: Yes, we agree to continue with DSM instead of PSM, the arguments of the Sydney group convinced us. We did try to define the various sources so readers can understand their purposes and methods.

6. The manuscript, combined with the discussion, will prompt many of us involved in soil survey to rethink about what is meant by soil classes versus soil properties, soil entities, taxonomic units or mapping units in the context of DSM. It may even cause many to re-visit the foundational mapping and taxonomy concepts of "soil body," "polypedon," and "soil individual."

A: Agreed. This discussion is outside the scope of this paper, and we hesitate to include it in our conclusions, as it opens a wide area which is better placed in a separate "concept" paper.

7. Several comments were made by the community and referee, which the authors acknowledge. I would add to those comments the need for making the figure captions (those actually attached to the figures) independently lucid.

A: RC2 also commented on this: "Most figure captions are more like titles. Please elaborate in the figure captions to guide the reader in what to look for in the figures."

We responded: "Our practice has been to point out the most important features of a figure in the text, at the point the figure is referenced. In some cases this text would be too long to fit into a caption. However have expanded most figure texts to make them independently understandable."
* * *
SOIL-2021-80 Reply to CC1

We thank this group from the University of Sydney for their perceptive comments. We reply with the same headings as their comments.

1 Digital Soil Mapping

Both these colleagues and one reviewer prefer the more common term "digital soil mapping" to "predictive soil mapping", making the argument that (1) DSM is the more common term, especially in recent highly-cited literature; (2) all maps are in some sense predictive. We accept this comment and will modify the text accordingly. We do still refer to the review by Scull, although his term "predictive" no longer appears in the text or title.

The commenters fairly summarize the main characteristics of the three DSM products; we feel this is well-explained already in the text.

**2 Soil Geography**

"The paper talks about soil geography. What might we mean by that? Generally, it can be taken to mean the spatial distribution of soil entities or the evolution of the spatial distribution of soil entities." Our meaning is the first, however, our "entities" have been represented by soil properties, as explained next.

In the revision we have made it clear that discretization of continuous maps to level sets was a first approach to comparing the patterns of continuous maps. Of the DSM approaches evaluated, none predict classes. POLARIS does, giving the predicted probability of many classes (soil series). Soil Properties and Class 100m Grids of the United States (SPCG) also predicts classes, but at the level of Soil Taxonomy Great Groups, which may be 100's of series. SoilGrids v2 does not (yet) predict classes, and when it does these will likely be Reference Groups, perhaps with single qualifiers, in the World Reference Base (WRB) system.

So, comparing these patterns is not yet possible. Indeed it is an important research area: "Description of the spatial distribution of soil classes remains an underdeveloped area of pedometrics." Since our classes are histogram-equalized reclassifications (i.e., ordinal classes), not nominal classes as in a soil classification system, we do not want to bring up this topic in this paper.

The commenters ask for "spatial methods that recognise continuity". As pointed out, we used the variogram, and the reduced sill due to regression is not relevant to our comparison – we are not claiming the sill represents the original variables, rather (as pointed out) the predicted variables.

**3 Ground Truth**

We did not intend to use SSURGO as actual ground truth, and indeed we point out the various reasons why it may not be accurate, even in the context of its design scale. We are sorry that the paper gave this impression. In the revision we have modified the text in various places to make this clear. Notice that the first Reviewer stated "I fully support the idea of using local soil maps elaborated by experienced soil surveyors as an alternative (complementary) ground-truth, despite the well-known weakness of soil maps" and this was indeed our intention.

We did not mean to imply that "the final goal of DSM is to recreate a polygon map". No, we are interested in digital soil maps (per grid cell). We have reviewed the text and adjusted it so as to remove this impression,

"The authors do not mention the intrinsic uncertainty of mapping units which are not homogeneous as a single polygon might suggest." We pointed out that polygons of SSURGO are linked to multiple constituents and estimates of their proportion.

"The North American mental model tends to focus most on soil topographic relationships whereas the digital soil mapping approaches are more explicitly multi-factorial." This is not the case. The USA model (we can not speak for Canada or Mexico) is explicitly multi-factorial, the main factors being (indeed) topography, but also vegetation/land use and geomorphic relations (e.g., post-glacial features, playas or alluvial fans), and soil surface features visible on airphotos, e.g., salinity. Field surveyors use all these clues to locate point observations and, especially, polygon boundaries.

We agree that "to do a convincing [pointwise] comparison, it is important to have an independently observed dataset with which to compare the various representations". This is

of course not feasible with our resources. We do have the point observations from NASIS but this is a heavily-biased sample set as it was mostly from purposive sampling of representative pedons, and of course was used in the model building for all three DSM products. If our aim were to compare maps by their success at reproducing points, we could have done statistical evaluation on this set. However, our objective, as stated in the Introduction, is to evaluate the spatial pattern and relation to soil geography. For this the "ground truth" can not be a set of points, it must be some pattern. Hence our use of SSURGO.

"Ideally such a comparison of the various maps with the independent observed dataset will be made in a statistically robust way, i.e. through the use of probability sampling and design-based inference." Indeed if the aim is to evaluate success of point-wise predictions. But this is not our aim, and we have tried to make the distinction clearer in the text.

We do know and use solar and Taylor diagrams for model evaluation, but again, these pointwise techniques are not applicable for our purpose.

4 The Way Forward

We agree that in many areas of the world with poor resources for systematic soil survey DSM is likely to be by far the most used method of making up-to-date soil maps. But we do not agree with this comment as it applies to the USA (our study area) nor other countries with active soil survey programmes. The position of the NRCS on this has been summarized as follows in the revised paper:

"In the USA (our study area) and in other countries with active soil survey programmes, DSM will be an important but not dominant tool in the overall survey. Soil survey as practiced by the NRCS uses methods from DSM, applied statistical modeling, and numerical ecology, along with an active and focused field programme. For example, supervised classification of terrain derivatives and satellite imagery has been successfully used to check internal consistency of map unit concepts and assist with the placement of delineations. The aim is to blend the most applicable tools from traditional field survey and applied statistical methods, supported by pedologic theory and regional land use considerations."
* * *
SOIL-2021-80 Reply to CC2

We appreciate the overall positive comments of the colleague and the appreciation for our efforts "attempting to shed some light on the touch points of inadequacy that exist in these products".

We agree that "distillation of examples would be useful with more contextual background to explain what the reader should be specifically looking for in the visual comparison outputs. Many articulate and powerful examples of where these products are lacking could be described for any given soilscape within the CONUS area". We decided in the main paper to concentrate on methods, and placed the four extensive case studies in the companion ISRIC report. These are not sufficient to "distill an overall evaluation over CONUS" as requested. Therefore we provide the code. We hope the work will be taken forward by others, especially within NRCS, to write a paper with this theme.

The paper states "This (the above attractions of PSM/DSM) removes the need for expertise in discovering and interpreting the soil-landscape relations, also known as the 'paradigm' of soil survey (Hudson, 1992), which is vital for traditional soil survey and difficult to acquire and harmonize among surveyors."; the colleague takes exception to this:

As one who regularly "actually examines the soil and landscape" of my area, I take great issue

with the above statement. It needs to be clarified that it is really only possible to evaluate the PSM/DSM results if one has the expertise derived from the traditional "paradigm" of soil survey.

We can see how our statement can be interpreted this way, and how incorrect it is from that point of view. We did not intend to imply that DSM replaces the paradigm. In the revision hope to have made this clear.

The colleague states "A dirty little secret concerning PSM/DSM products is that in spite of their assumed superiority and 'explicitly multifactorial' approach as previously described by esteemed commentators, these products are not 'intelligent' in how they parse soil topographic/soil geomorphic/soil geographic relationships."

We are not sure how much of a secret this is, at least to us, and we certainly are not the "esteemed commenters"! It may be that DSM has been so presented and thus over-sold. Again, in the revision havel highlighted this inherent limitation of DSM.

The colleague states: "No matter how you slice it understanding local geomorphic relationships and their cut in the process of mapping soils. These complex soil geomorphic relationships live in conventional soil survey products and are largely absent in PSM/DSM products making them substandard in how they capture the reality of soil distributions. You pick the area (any area) and a soil scientist with the ability to read and understand soil geomorphic relationships (the quote traditional paradigm of soil survey) will show you numerous shortcomings."

The subsequent two paragraphs expand on this point.

We completely agree and had thought to have made this clear. In fact the motivation for this paper was exactly to highlight this point. Since DSM is so popular it should be critically examined, as we have tried to do. In the revision we hope to have ensured that this point is well brought out.

The colleague states: "The paper should postulate on why the PSM/DSM products don't measure up to conventional survey. The intent is not to replicate earlier products, but one would hope that PSM/DSM products would reflect a similar lineage in larger structures and spatial patterns expressed within the soilscape. We know these patterns are there so why do we deny them in these new products."

Indeed. We had hoped that the spatial patterns we know from expert soil-landscape analysis would be reflected in the DSM products. As the paper and case studies show, there are serious deficiencies in the examined products, and we expect in all products made by similar methods.

We had not postulated on the reasons. We appreciate the stimulus given by the commentator to do so, and we have added discussion of this issue in the Conclusions. These possible reasons have been added to the Conclusion:

-- The dominant DSM methods do not explicitly consider spatial continuity or pattern. Experiments have been started with convolutional neural networks and other methods with varying window sizes of covariates.

-- Environmental covariates to represent past soil-forming conditions (the "time" factor) are only available since the satellite remote sensing age, very short in terms of soil formation.

-- Point observations are mostly placed at "typical" or "representative" locations and do not

capture the full range of variability along toposequences.

-- Poor georeference of legacy point observations leads to poor correlation with environmental covariates, hence to poor models, hence to much noise in the DSM product, which can obscure patterns.

6. Finally, the colleague states "I think the more pressing and important question is how do we build the intelligence and paradigm from traditional mapping into PSM/DSM approaches so that the strengths of the hierarchical relationships of geomorphology, superposition, fluvial downcutting/cross-cutting, geologic discontinuities of materials are added back into these models to further inform the outputs."

We completely agree, and, along with others, are active in attempting to develop such methods.

It should be noted that the "SolIM" approach of Zhu and colleagues already in 1997 took an expert-based approach to DSM. This is applicable in small areas with detailed knowledge of the soil-landscape relations, but not to wide-area models. This approach still needs covariates for the model, and if these do not cover the soil-forming environment, it will also have difficulty.

Reference: Zhu, A.-X., Band, L. E., Vertessy, R., & Dutton, B. (1997). Derivation of soil properties using a soil land inference model (SoLIM). Soil Science Society of America Journal, 61(2), 523-533.

The work in this paper has been improved and applied in many further works mostly by Prof. Zhu's groups in Madison, Beijing and Nanjing.
* * *
SOIL-2021-80 Reply to CC3

1 General comment

Comment: "My only critical comment in the pdf concerns what appears to be a mis-characterization of POLARIS uncertainty estimates. Otherwise the paper is well written and provides a thoughtful comparison on the differences amongst soil datasets."

Answer: The paper states "SG2 and PSP predict the 5% and 95% quantiles of the distribution of predictions using Quantile Regression Forests (QRF)".

Indeed this is not correct for PSP. It does provide these quantiles but using a more complex method, as described in §3.3.1 of the POLARIS soil properties paper (DOI:10.1029/2018WR022797). It is based on property data available for a given soil series, to create a depth-harmonized profile with uncertainty at each depth slice. We have changed L242-3 accordingly. The important point for our paper is that the PSP method of DSM provides uncertainty, which we can compare with other measures of uncertainty, in this case with SG2.

2 Specific comments

Line 60: Does it? Domain expertise seems critical to know when these ML models are overfit.

Answer: this refers to the statement "This removes the need for expertise in discovering and interpreting the soil-landscape relations,..." What we meant here is that expertise is not needed in the same sense as the surveyor uses holistic, expert knowledge of the soillandscape relation. Indeed domain expertise should be used to (1) select relevant covariates, (2) check that the ML model output is reasonable.

In the revision we have clarified this point: "However, expertise in soil-landscape relations is still needed to ensure that DSM outputs are reasonable, and to discover reasons for any discrepancies."

Line 242: PSP used a "classification" forest, thus how could it has also used a quantile "regression" forest? The PSP is based on a weighted average of soil components.

Answer: Correct, we answered this in the general comment above.

Line 405 / Table 1: What is with the units in this table? The RMSD is different by a pH of 4–6? You're multiplying by 10? Why?

Answer: Because of processing limitations, it is common in wide-area (e.g., global) DSM models to predict integer values. This is to reduce the size of the generated raster layers, using one-bit signed integers (range 0–255). Thus a single-precision pH 4.2 is represented as 42, etc. We created the table from these integers, to be consistent with the necessary use of integers in the raster maps, e.g., Fig. 4. We also decided that, given the imprecision of the ML models, a single-precision pH was sufficiently precise. We made similar decision in SoilGrids v2.0 for each property, as shown in the Case Studies ISRIC report.

Figure 3: I love visuals, but in this case a simple correlation matrix would be more informative.

Answer: This plot is quite common in related literature, and the exact values are of less interest (we think) than the visual impression given by the plot.

Table 2: I would have figured that gNATSGO would have had the highest nugget because it is the most detailed, but I suppose this is an artifact of the polygons.

Answer: We think this is indeed an effect of the polygons: adjacent grid cells are more likely to be within a polygon than between them. Notice also the short range, which reflects the typical width of a delineation.

Figure 11: Red-Green is a bad color scheme for folks (like me) whose vision is color deficient.

Answer: Yes, and we were thoughtless here. For the revised paper we have recompiled the figures, after consulting with experts in colour schemes appropriate for red-green colour "blindness".

Figure 12: Any thoughts on the use of a consistent color scheme across all thematic maps?

Answer: We purposely used different schemes for (1) property predictions, (2) property prediction differences, (3) uncertainty, (4) uncertainty differences (IQR), (5) classified maps – the red/green problem mentioned above, although here we were not consistent in the regional vs. local class maps. We have rectified this last, along with the previous point, in the revised paper.

Figure 13: Shouldn't you have a legend for the gridded soil maps? Also, IMO I find the transparency and orientation distracting.

Answer: Here we were just trying to show the difference between gSSURGO and PSP, i.e., PSP's disaggregation, against a landscape with significant relief and composite map units.

The actual values are not so important, the reader should concentrate on matching the colours, which do represent the same values.

The transparency and orientation are to highlight the landscape underneath the soil map.